# SCALING EVALUATION-TIME COMPUTE WITH REASONING MODELS AS PROCESS EVALUATORS

## ABSTRACT

Language model (LM) evaluators that generate chain-of-thought (CoT) reasoning are widely used for the assessment of LM responses. Simultaneously, increasing LMs' "thinking" time through scaling test-time compute has proven to be an effective technique for solving challenging problems in domains such as math and code. This raises a natural question: can an LM's evaluation capability also be improved by scaling test-time compute? To answer this, we investigate employing *reasoning models* – LMs that natively generate long CoT reasoning – as evaluators. We explore scaling evaluation-time compute by using reasoning models to evaluate both the overall candidate response (*i.e.*, outcome evaluation) and the individual reasoning steps within it (*i.e.*, process evaluation). In our experiments, we observe that evaluator performance improves monotonically with the number of reasoning tokens generated, mirroring trends seen in LM reasoning. Furthermore, we use these more accurate evaluators to rerank multiple generations, and demonstrate that spending more compute at evaluation time can be as effective as increasing compute during generation for improving an LM's problem-solving performance.

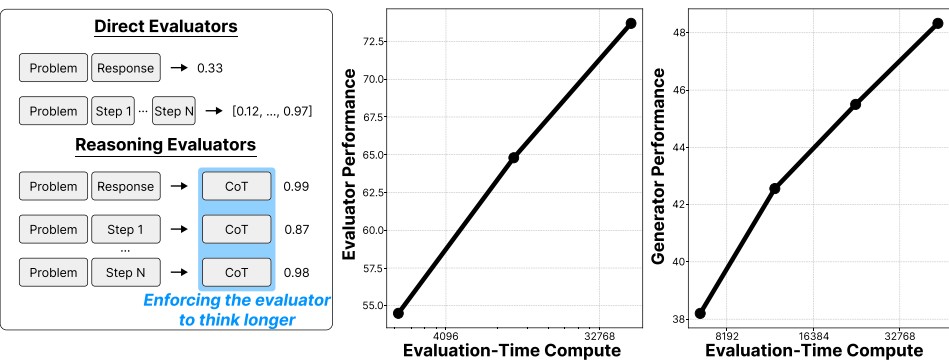

Figure 1: **We investigate the effect of scaling test-time compute for evaluation (evaluation-time compute)**: Enforcing the generation of additional reasoning tokens leads to improved evaluation performance (section 3). This can, in turn, be utilized to further improve the generator's problem-solving capabilities (section 4).

## 1 INTRODUCTION

Research on language models (LMs) involves an interplay between generation and evaluation: better generators require better evaluators and better evaluators can further enhance generators. For instance, an evaluator can verify the quality of the generator's response (Liang et al., 2023; Zheng et al., 2024b; Ye et al., 2024) or identify the parts of a generator's response that contain mistakes (Lightman et al., 2024; Zheng et al., 2024a; Zhang et al., 2025). Furthermore, the generator's performance can be improved by integrating the better evaluator into its inference-time algorithms (Cobbe et al., 2021; Uesato et al., 2022; Lightman et al., 2024; Sun et al., 2024; Wu et al., 2024a).

Reasoning models have opened up a new paradigm for generation based on generating a long chain-of-thought (CoT) (Jaech et al., 2024; Guo et al., 2025; Muennighoff et al., 2025; Aggarwal and Welleck, 2025). Prior works have found that generating long CoTs is an effective strategy for leveraging

test-time compute to solve difficult tasks that conventional instruction-tuned models cannot (Yeo et al., 2025; Xu et al., 2025). However, it is unclear whether evaluators, like generators, can also be improved by scaling test-time compute with long CoTs. In this paper, we ask and answer two questions: (1) Can we replicate test-time scaling behavior observed in generators with evaluators? And if so, (2) can this improved evaluation ability further improve generation results as well?

Our main contribution is an examination of generative evaluators that use long CoT reasoning, which we refer to as *reasoning evaluators*. Reasoning evaluators are obtained by prompting reasoning models – a class of models trained to produce long CoT either using RL or through distillation from the outputs of such a model (Guo et al., 2025; Team, 2024b) – to act as evaluators. By generating long outputs exhibiting complex reasoning patterns including self-verification and backtracking (Gandhi et al., 2025b; Lu et al., 2025), reasoning models expend test-time compute to attain improved reasoning capabilities, a trait that we hypothesize also makes them better evaluators. We contrast reasoning evaluators with (1) *direct evaluators*, which predict scores without CoT reasoning (Lightman et al., 2024; Cobbe et al., 2021), and (2) *fine-tuned generative evaluators* that produce shorter CoT lacking the complex reasoning patterns found in reasoning model outputs (Zhang et al., 2024; Ankner et al., 2024; Kim et al., 2024a). As shown in the left of Figure 1, we force reasoning evaluators to generate more reasoning tokens by prompting them to evaluate both each step of an output individually as well as the solution as a whole. This recipe unifies techniques from prior work on step-by-step evaluation (process reward models; PRMs; (Lightman et al., 2024; Wang et al., 2024a; Zheng et al., 2024a)) and outcome-based evaluation (outcome reward models; ORMs; (Cobbe et al., 2021; Liu et al., 2024)).

We demonstrate the effectiveness of our approach across two settings. First, as shown in the middle plot of Figure 1, we show that the evaluator's performance improves monotonically as it generates more reasoning tokens. We further show that a 32B reasoning evaluator can outperform a 72B state-of-the-art PRM by a 4.5% margin on ProcessBench (Zheng et al., 2024a), a benchmark that measures whether an LM can identify the first occurring error within a given response. This is notable because, while existing direct evaluators are trained on extensive process supervision, our reasoning evaluators achieve strong performance through *evaluation-time scaling* alone, without training.

Second, as shown in the right plot of Figure 1, we find that evaluation-time scaling is an effective method for further improving the generator's performance. When integrating reasoning evaluators into Best-of-$N$ sampling, where an evaluator reranks multiple solutions sampled by a generator, our reasoning evaluators using Best-of-8 outperform direct evaluators (*e.g.*, ORMs, PRMs) using Best-of-64 by a 4.30% to 6.63% margin given a fixed compute budget, highlighting the potential benefits of spending more test-time compute for evaluation at the expense of sampling more responses.

## 2 METHODOLOGY

We describe our approach for scaling evaluation-time compute by assessing both overall responses (subsection 2.1) and individual response segments (subsection 2.2) with reasoning evaluators. We then explain how we combine process and outcome judgments (subsection 2.3) for further gains.

**Reasoning Evaluators vs. Direct Evaluators** We refer to conventional evaluators that are trained to map a problem and a response (or steps) to a scalar value score as *direct evaluators*. Reasoning evaluators differ from direct evaluators in two aspects. First, reasoning evaluators generate chain-of-thought (CoT) reasoning before predicting the final judgment. Second, while direct evaluators with a specialized reward modeling head must be fine-tuned, reasoning evaluators may either be specifically trained for evaluation [1] or may be off-the-shelf LMs that are prompted to act as evaluators. In this paper, we focus on the latter approach by prompting reasoning models to function as evaluators.

Given a problem $x_i$ and response $y_i$, the evaluator is used to estimate the "goodness" of $y_i$ by generating a score $s_i$. This score can be obtained using a trained reward modeling head (direct evaluators) or from the logits of answer tokens (*e.g.*, 0/1) (reasoning evaluators). The mapping function of an outcome and process evaluator can be expressed as:

- **Outcome Evaluator**: $(x_i, y_i) \rightarrow s_i$

---

[1] We also classify trained evaluators such as CLoud (Ankner et al., 2024) and Prometheus (Kim et al., 2024c) as reasoning evaluators, although they produce relatively short CoTs that lack the versatile reasoning patterns seen in reasoning models outputs.

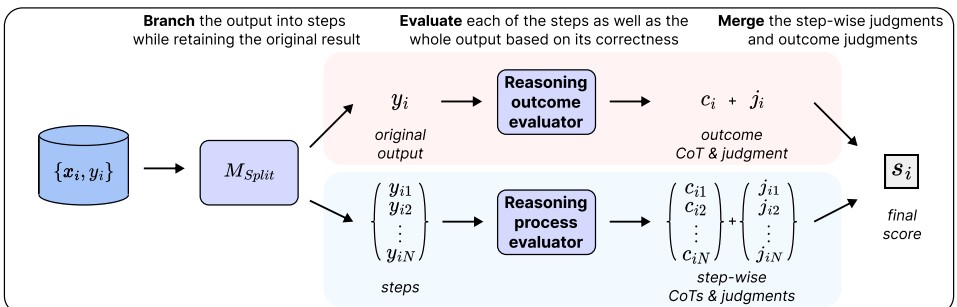

Figure 2: We propose scaling evaluation-time compute by using the evaluator to assess both the overall output (light red) and its constituent reasoning steps (light blue) and then combining the judgments into a final score.

- **Process Evaluator**: $(x_i, [y_{i1}, y_{i2}, ..., y_{iN}]) \rightarrow [s_{i1}, s_{i2}, ..., s_{iN}]$.

Process evaluators require a **splitting function** to divide $y_i$ into discrete steps $[y_{i1}, y_{i2}, ..., y_{iN}]$. Furthermore, process evaluators can only be used for evaluation of the final outcome if provided with an **aggregation function** that maps per-step scores $[s_{i1}, s_{i2}, ..., s_{iN}]$ to an aggregated final score $s_i$. Conventionally, a heuristic-based approach is used as the splitting function (*e.g.*, splitting based on "\n\n") while the min function ($s_i = min(s_{i1}, s_{i2}, ..., s_{iN})$) is often used as the aggregation function (Lightman et al., 2024; Wang et al., 2024a; Sun et al., 2024). As discussed in the previous paragraph, direct outcome and process evaluators predict these values through specially trained heads. In the following subsections we discuss how to make these predictions with reasoning models.

## 2.1 REASONING OUTCOME EVALUATORS

Our reasoning outcome evaluators have at their core a function

$$(x_i, y_i) \rightarrow (c_i, j_i), \tag{1}$$

where $c_i$ denotes a CoT, and $j_i$ is the evaluator's judgment, represented as the probability distribution over tokens in the vocabulary: see the upper section of Figure 2 (colored light red). We prompt the LM to output "1" if the response is deemed to be correct and "0" if not. To transform $j_i$ into a scalar value score $s_i$, we use the logits $\ell$ of "1" and "0" tokens and perform a softmax operation:

$$s_i = \frac{e^{\ell(j_i=1)}}{e^{\ell(j_i=0)} + e^{\ell(j_i=1)}}. \tag{2}$$

## 2.2 REASONING PROCESS EVALUATORS

We formulate the mapping function for assessing reasoning step $k$ as

$$(x_i, [y_{i1}, ..., y_{ik}]) \rightarrow (c_{ik}, j_{ik}) \quad (1 \leq k \leq N), \tag{3}$$

where $c_{ik}$ denotes the CoT that examines $y_{ik}$ for potential logical flaws or inconsistencies and $j_{ik}$ denotes the judgment for $y_{ik}$, which is also represented as a probability distribution: see the bottom section of Figure 2 (colored light blue). Note that the previous steps $[y_{i1}, ..., y_{i(k-1)}]$ are provided as context for precise assessment of the current step $y_{ik}$. Then, to convert $j_{ik}$ into $s_{ik}$, we use

$$[s_{i1}, ..., s_{iN}] = \left[ \frac{e^{\ell(j_{i1}=1)}}{e^{\ell(j_{i1}=0)} + e^{\ell(j_{i1}=1)}}, ..., \frac{e^{\ell(j_{iN}=1)}}{e^{\ell(j_{iN}=0)} + e^{\ell(j_{iN}=1)}} \right]. \tag{4}$$

**Single- vs. multi-step process evaluation** The formulation above forms the core of our proposed method, but we also compare with an ablated *single-step* reasoning process evaluator proposed by Zheng et al. (2024a) that generates a single CoT before making judgments of all $N$ steps in $y_i$ as

$$(x_i, [y_{i1}, ..., y_{iN}]) \rightarrow (c_i, [j_{i1}, ..., j_{iN}]). \tag{5}$$

Unless explicitly stated, reasoning process evaluation refers to our proposed multi-step variant. Evaluating each step *separately* is our preferred method for two reasons: (1) evaluating all $N$ steps at

once risks exceeding the context window of the reasoning models, and (2) stepwise evaluation forces the evaluator to assess each step more thoroughly, thereby naturally scaling evaluation-time compute.

**Choice of splitting function and aggregation function**     Additionally, we make the following adjustments to the splitting and aggregation functions when using reasoning process evaluators:

- **Model-based splitting**: When splitting $y_i$ into $[y_{i1}, ..., y_{iN}]$, conventional heuristic-based approaches may be ineffective for some cases (*e.g.*, when $y_i$ does not include "\n\n" or is not written in a structured format, as is the case for code). To deal with this, we adopt model-based splitting where an LM $M_{split}$ is prompted to insert an indicator phrase "[SPLIT]" between steps:

$$M_{split} : y_i \rightarrow [y_{i1} \texttt{ [SPLIT] } y_{i2} \texttt{ [SPLIT] } ... \texttt{ [SPLIT] } y_{iN}]. \tag{6}$$

- **Score aggregation**: After acquiring $[s_{i1}, ..., s_{iN}]$ as in Equation 4, we aggregate the $N$ judgments into a single scalar value score $s_i$. In our experiments, we find that the `mean_logit` function (Sun et al., 2024) yields better results than `min` function. The `mean_logit` function is expressed as

$$s_i = \texttt{mean\_logit}(s_{ik}) = \sigma \left( \frac{\Sigma_k \frac{s_{ik}}{1 - s_{ik}}}{N} \right) \ \ (1 \leq k \leq N). \tag{7}$$

Note that these adjustments can be applied to direct process evaluators as well: see Appendix C.

## 2.3 Combining outcome judgments and process scores

While the objective of outcome evaluation is to determine the *correctness of the final answer*, the objective of process evaluation is to determine the *correctness of each step*. Both have their advantages in identifying reasoning errors – outcome evaluation takes a more holistic approach while process evaluation can potentially identify more fine-grained errors. Inspired by Uesato et al. (2022), we consider combining both outcome and process evaluation scores through interpolation as follows

$$s_{final} = \alpha \cdot s_{outcome} + (1 - \alpha) \cdot s_{process}. \tag{8}$$

Here, choosing $\alpha = 0$ is identical to only using the process score and choosing $\alpha = 1$ is identical to using only the outcome score. We use $\alpha = 0.5$ to avoid overfitting to either approach and refer to this method as **reasoning process + outcome evaluation**. See Appendix I for analyses on varying $\alpha$.

## 3 Evaluation-Time Scaling Trends for Process Evaluation

As discussed in section 2, our approach involves assessing output steps *individually* (multi-step process evaluation) using reasoning evaluators. To examine the effectiveness of this choice, we experiment on a response error detection task and compare our method against state-of-the-art PRMs.

### 3.1 Experimental setting

**Benchmark**     We explore scaling evaluation-time compute with ProcessBench (Zheng et al., 2024a), which includes diverse responses from different LMs and highly reliable human-annotated labels. In ProcessBench, evaluators are tasked with identifying the first paragraph in the solution that contains incorrect logic, if any. ProcessBench has 3,400 instances, with queries sampled from 4 different math benchmarks and responses from 12 distinct LMs. Each response consists of 7.56 steps on average.

**Metric**     The evaluator's performance on ProcessBench is measured using the F1 score, computed from the precision and recall of predicting the index of the first paragraph that contains a logical error: evaluators are penalized for incorrectly identifying a paragraph as erroneous when no error exists, misidentifying the index of the erroneous paragraph, or failing to detect an error when one is present. If the evaluator predicts that one or more steps are incorrect, we use the earliest incorrect step (the step with the smallest index) as the final prediction, following Zheng et al. (2024a).

**Methods**     We consider the following evaluator baselines (see Appendix A.1 for the list of models):

- **Direct process evaluator**: We employ process reward models (PRMs) as direct process evaluators. Note that these models *do not* generate CoTs but instead directly predict the correctness of all steps.

Table 1: Experimental results for ProcessBench experiments. We find that evaluation performance can be improved by scaling evaluation-time compute using multi-step process evaluation and self-consistency. Notably, we find that even LMs not specifically fine-tuned for evaluation can outperform specially trained PRMs. $\Delta$ denotes gains associated with either applying (1) multi-step process evaluation or (2) self-consistency for the same backbone LM. The best performance within each category is **bolded**. See Table 4 for additional results.

| Model | GSM8K | MATH | Olym. Bench | Omni-MATH | Avg. F1 | $\Delta$ |
|---|---|---|---|---|---|---|
| **Direct Process Evaluator (PRMs)** | | | | | | |
| Qwen2.5-Math-7B-PRM800K | 68.4 | 62.5 | 50.4 | 43.6 | 56.2 | - |
| Qwen2.5-Math-PRM-7B | 82.4 | 77.6 | 67.5 | 66.3 | 73.5 | - |
| Qwen2.5-Math-PRM-72B | **87.3** | **80.6** | **74.3** | **71.1** | **78.3** | - |
| **Single-step Reasoning Process Evaluator** | | | | | | |
| **Instruction-tuned Models** | | | | | | |
| Llama-3.1-8B-Instruct | 24.0 | 15.5 | 9.7 | 10.1 | 14.8 | 0.0 |
| Qwen2.5-32B-Instruct | 63.8 | 47.5 | 35.9 | 32.7 | 45.0 | 0.0 |
| Qwen2.5-72B-Instruct | 76.2 | 61.8 | 54.6 | 52.2 | 61.2 | - |
| **Reasoning Models** | | | | | | |
| DeepSeek-R1-Distill-Qwen-7B | 68.3 | 61.1 | 48.3 | 40.4 | 54.5 | 0.0 |
| DeepSeek-R1-Distill-Qwen-32B | 83.9 | 78.1 | 72.4 | 67.7 | 75.5 | 0.0 |
| QwQ-32B-Preview | 77.5 | 58.9 | 31.2 | 35.8 | 50.9 | 0.0 |
| QwQ-32B | 79.5 | 77.5 | 71.5 | 69.4 | 74.5 | 0.0 |
| **Reasoning Models** (Self-Consistency) | | | | | | |
| DeepSeek-R1-Distill-Qwen-7B | 69.3 | 67.9 | 54.8 | 51.5 | 60.9 | +6.4 |
| DeepSeek-R1-Distill-Qwen-32B | 82.2 | **80.4** | **76.2** | 72.5 | **77.8** | +2.3 |
| QwQ-32B-Preview | **88.0** | 78.7 | 57.8 | 61.3 | 71.5 | +20.6 |
| QwQ-32B | 81.0 | 78.8 | 74.4 | **72.8** | 76.8 | +2.3 |
| **Reasoning Process Evaluator (Ours)** | | | | | | |
| **Instruction-tuned Models** (Multi-step Process Evaluation) | | | | | | |
| Llama-3.1-8B-Instruct | 35.2 | 22.8 | 12.6 | 17.9 | 22.1 | +7.3 |
| Qwen2.5-32B-Instruct | 70.1 | 61.7 | 54.2 | 53.9 | 60.0 | +15.0 |
| **Reasoning Models** (Multi-step Process Evaluation) | | | | | | |
| DeepSeek-R1-Distill-Qwen-7B | 75.5 | 67.3 | 59.8 | 56.6 | 64.8 | +10.3 |
| DeepSeek-R1-Distill-Qwen-32B | 80.3 | 82.2 | 77.0 | 75.0 | 78.6 | +3.1 |
| QwQ-32B-Preview | 81.7 | 79.3 | 70.3 | 69.8 | 75.3 | +24.4 |
| QwQ-32B | 81.5 | 83.6 | 76.8 | 75.1 | 79.3 | +4.8 |
| **Reasoning Models** (Multi-step Process Evaluation + Self-Consistency) | | | | | | |
| DeepSeek-R1-Distill-Qwen-7B | 80.1 | 75.1 | 69.3 | 70.4 | 73.7 | +19.2 |
| DeepSeek-R1-Distill-Qwen-32B | 86.6 | 85.4 | 78.9 | **80.3** | **82.8** | +7.3 |
| QwQ-32B-Preview | **86.8** | **85.7** | 79.0 | 78.0 | 82.4 | +31.5 |
| QwQ-32B | 85.0 | 85.6 | **79.4** | 78.0 | 82.0 | +7.5 |

- **Single-step reasoning process evaluator**: We adopt the approach proposed by Zheng et al. (2024a) and Zhang et al. (2025), where a language model is provided with the response and prompted to produce a single CoT as well as the index of the first paragraph containing a logical error, if one exists. This corresponds to the ablated "single-step" evaluator discussed in subsection 2.2.

- **Reasoning process evaluator**: We explore our approach for using reasoning models as process evaluators. This involves assessing each segment *individually* and determining its correctness. Note that for this experiment, we use a simplified version of the approach described in subsection 2.2: (1) a splitting function is not required as segments are already provided and (2) an aggregation function is not required as the goal of the benchmark is to predict the index of the segment containing the first mistake, not to produce an outcome score to compare multiple responses.

**Matching test-time budget across methods**  To ensure a fair comparison between single-step evaluators and our method under similar test-time compute constraints, we also experiment with **self-consistency** (Wang et al., 2023) for evaluation. Specifically, the evaluator generates $M$ CoT trajectories (*e.g.*, if applied to our reasoning process evaluator, it assesses each of $N$ steps $M$ times, resulting in a total of $N \cdot M$ inference calls), with $j_{ik}$ chosen based on a majority vote. Self-consistency is inapplicable for PRMs since they produce identical scores across multiple inference calls.

## 3.2 EXPERIMENTAL RESULTS

Our main results on ProcessBench are presented in Table 1. See Table 4 for additional results.

**Finding 1: Reasoning models are better evaluators than instruction-tuned models**. For example, DeepSeek-R1-Distill-Qwen-32B achieves an average F1 score of 75.5 when employed as a reasoning process evaluator, significantly outperforming the larger Qwen2.5-72B-Instruct model (61.2 F1) despite having only 44% as many parameters. This suggests that reasoning capability is associated with improved evaluation capability, even when the reasoning model is not trained as an evaluator.

**Finding 2: Single-step methods fail to match the performance of direct process evaluators**. For example, DeepSeek-R1-Distill-Qwen-7B achieves an F1 score of 54.5, which is lower than Qwen2.5-Math-7B-PRM800K (56.2) and Qwen2.5-Math-PRM-7B (73.5). Similarly, DeepSeek-R1-Distill-Qwen-32B (75.5) and QwQ-32B (74.5) fall behind the Qwen2.5-Math-PRM-72B (78.3). This indicates that single-step evaluation is not sufficient to identify errors in responses.

**Finding 3: It is important to evaluate each step individually (multi-step process evaluation)**. As detailed in the previous subsection, we compare two approaches for scaling evaluation-time compute: (1) self-consistency, where the evaluator generates multiple CoTs for evaluation and the final judgment is decided via majority vote and (2) multi-step process evaluation, where the evaluator assesses each step in the solution individually. As ProcessBench has an average of 7.56 steps and we generate 8 CoTs for the self-consistency baseline, the two approaches incur similar inference costs.

Our results indicate that prompting reasoning models to evaluate each step individually is more effective than applying self-consistency given a fixed evaluation compute budget. Using multi-step process evaluation, DeepSeek-R1-Distill-Qwen-7,32B, QwQ-32B-Preview, and QwQ-32B achieve F1 scores of 64.8, 78.6, 75.3, and 79.3, respectively. In contrast, applying self-consistency yields lower scores of 60.9, 77.8, 71.5, and 76.8. Notably, our multi-step process evaluation method enables DeepSeek-R1-Distill-Qwen-32B (78.6) and QwQ-32B (79.3) to outperform Qwen2.5-Math-PRM-72B (78.3) – a model nearly twice as big – without any additional training. Furthermore, instruction-tuned models also achieve sizable gains when used for multi-step process evaluation, with Llama-3.1-8B-Instruct and Qwen2.5-32B-Instruct demonstrating gains of +7.3 and +15.0 respectively. These results indicate that our method can be applied to non-reasoning LMs as well.

**Finding 4: Combining self-consistency and multi-step process evaluation can further enhance performance**. We also find that further scaling evaluation-time compute by applying multi-step process evaluation and self-consistency together yields even more gains. For instance, DeepSeek-R1-Distill-Qwen-7B (73.7) outperforms Qwen2.5-Math-PRM-7B (73.5). Similarly, DeepSeek-R1-Distill-Qwen-32B (82.8) and QwQ-32B (82.0) surpass Qwen2.5-Math-PRM-72B (78.3), which was the previous state-of-the-art (Zhang et al., 2025), suggesting that both self-consistency and multi-step process evaluation can provide complementary benefits for scaling evaluation-time compute. This is notable because the reasoning models are not explicitly trained as evaluators, yet increasing evaluation-time compute consistently improves performance and achieves state-of-the-art results.

## 4 TRANSLATING IMPROVED EVALUATION TO PROBLEM-SOLVING

Building upon the previous section, we explore whether we can leverage evaluation-time scaling to improve the generator's performance. We investigate this with Best-of-$N$ sampling, where test-time scaling is conventionally achieved by generating more response candidates. In contrast, we explore whether allocating more evaluation-time compute at the expense of sampling fewer candidate responses can ultimately yield better performance given a fixed test-time compute budget.

### 4.1 EXPERIMENTAL SETTING

**Benchmarks** We follow the setting of Cui et al. (2025): we utilize three LMs as generators, namely Eurus-2-SFT (Cui et al., 2025), Llama3.1-70B-Instruct (Llama Team, 2024), and Qwen2.5-7B-Instruct (Yang et al., 2024). Using our generator LMs, we generate 64 responses per instance across seven benchmarks (AIME24, AMC23, Minerva Math (Lewkowycz et al., 2022), OlympiadBench (He et al., 2024), MATH500 (Hendrycks et al., 2021), LeetCode (Guo et al., 2024), and GPQA (Rein et al.,

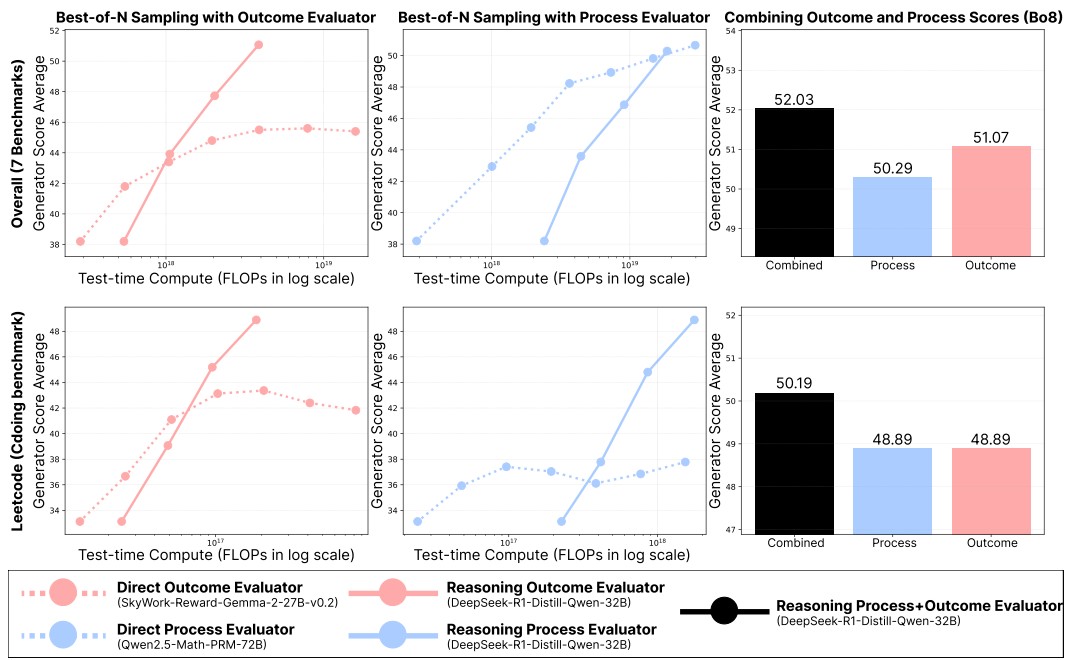

Figure 3: Best-of-$N$ experimental results for 27B $\sim$ 72B scale evaluators. Test-time compute on the x axis denotes the generator's test-time compute added with evaluation-time compute. **(left, middle)** We compare direct evaluators (using Best-of-64) against reasoning evaluators (using Best-of-8), with each dot representing a doubling of the number of responses. We find that reasoning evaluators achieve better performance compared to their direct counterparts given a fixed test-time compute budget: see Appendix E for details of our FLOPs calculations and Table 2 for full results. **(right)** Combining outcome and process scores yields further gains.

2024)), covering a total of 4,680 instances and 299,520 responses. In Best-of-$N$ setting, evaluators assess and rank the $N$ responses, with the highest-scoring response chosen as the final prediction.

**Metrics** For LeetCode, we report pass@1, which measures whether a response passes all test cases. For the remaining 6 benchmarks, we report accuracy scores, which measures whether a response is correct. We report the average score of the 3 generators across all 7 benchmarks (21 settings in total).

**Methods** We consider the following evaluator baselines (see Appendix A.2 for the list of models):

- **Direct outcome evaluator**: We employ outcome reward models (ORMs) as direct outcome evaluators. Note that these models do not generate CoTs but instead directly predict the scores.

- **Direct process evaluator**: We employ process reward models (PRMs) as direct process evaluators. We adopt the splitting functions and aggregation strategies accompanying each PRM's official implementation, as specified in their GitHub repositories or Hugging Face model cards. Note that these models do not generate CoTs but instead directly predict the correctness of all steps.

- **Reasoning outcome evaluator**: We prompt reasoning models to act as reasoning outcome evaluators. The evaluator first generates a CoT, followed by a judgment for the correctness of the overall response. Details are included in subsection 2.1.

- **Reasoning process evaluator**: We prompt reasoning models to act as reasoning process evaluators. The evaluator individually generates a CoT as a judgment for each step. We use our proposed model-based splitting strategy and the `mean_logits` aggregation function (see subsection 2.2).

- **Reasoning process + outcome evaluator**: Using the judgments from a reasoning outcome evaluator and a reasoning process evaluator, we combine scores to obtain the overall score for the response. Details are included in subsection 2.3. We use $\alpha = 0.5$ to avoid overfitting to the test set.

In Appendix A.2, we experiment with generative evaluators that were fine-tuned to function as outcome evaluators, and also study the effects of applying model-based splitting and `mean_logits` aggregation to direct process evaluators. Prompt templates can be found in Appendix K.

**Matching evaluation-time budget across baselines**    We test direct outcome evaluators and direct process evaluators in the Best-of-64 setting. To account for the higher per-instance inference cost of reasoning evaluators, we evaluate them in the Best-of-8 setting instead, thereby ensuring a compute budget comparable to that of direct evaluators. We use `Qwen2.5-72B-Instruct` to segment the response into steps, yielding 10.07 steps per response on average across the 21 settings.

## 4.2 EXPERIMENTAL RESULTS

**Finding 1: Scaling evaluation-time compute is more effective than generating additional candidate responses given a fixed compute budget**. From the top left and top middle plots in Figure 3, we observe that reasoning evaluators using Best-of-8 – including both process and outcome evaluators – achieve performance equal to or better than direct evaluators using Best-of-64, while requiring similar or less test-time compute. Looking at the overall scaling trends, we find that reasoning evaluators suffer less from *reward model over-optimization*, a phenomenon where Best-of-$N$ gains plateau or diminish with increased $N$ (Gao et al., 2023; Rafailov et al., 2024). Reward model over-optimization arises from imperfections in the reward function's approximation of the ground truth (Lambert and Calandra, 2023); our results suggest that scaling evaluation-time compute can partially mitigate this issue by providing more robust evaluation. Similar trends are also observed in smaller evaluators (7B): see Figure 5 in Appendix C.

**Finding 2: Combining scores from reasoning outcome and process evaluation can boost performance**. From the top right plot of Figure 3, we observe that combining outcome and process scores can yield improved results. We hypothesize that the two evaluation approaches provide *complementary* signals that enable more accurate assessment. We investigate this hypothesis in Appendix H.2 and also further study the effects of altering $\alpha$ in Appendix I.

**Finding 3:  Reasoning evaluators are especially effective for assessing code outputs**. From the bottom left and bottom middle plot in Figure 3, we observe that reasoning evaluators not only significantly outperform direct evaluators while using less test-time compute, they also exhibit promising scaling trends when assessing code outputs. On the other hand, both the direct outcome evaluator and direct process evaluator heavily suffer from reward model over-optimization. We attribute these findings to two main causes. Firstly, direct process evaluators are often only trained on math data (*e.g.*, PRM800K (Lightman et al., 2024)), making them less effective on out-of-domain tasks such as coding. Secondly, the heuristic-based splitting methods (*e.g.*, splitting based on newline characters) typically adopted for direct process evaluation may be suboptimal for code outputs.

**Additional results in Appendix**. In Appendix C, we discuss (1) the scaling trends of 7B-sized evaluators, (2) how single-step process evaluators and fine-tuned reasoning evaluators (Ankner et al., 2024; Kim et al., 2024a) compare to our baselines, and (3) the effects of applying model-based splitting and different aggregation functions. Furthermore, we explore evaluating reasoning model-generated traces using reasoning models (self-evaluation) in Appendix G.

## 5 ANALYSES ON WHY REASONING PROCESS EVALUATORS ARE EFFECTIVE

**Why are reasoning evaluators effective?**    The top row of Figure 4 presents descriptions of common patterns observed in the multi-step process evaluation outputs of reasoning evaluators, accompanied by representative examples from evaluations on a LeetCode problem. It is known that reasoning models use versatile reasoning patterns when solving problems (Gandhi et al., 2025a), such as generating "Wait" tokens that encourage verification between solution steps; similar patterns are also observed when reasoning models are prompted to function as evaluators. Some patterns we observe in our example include (1) examining what implementation the given step contains, (2) conducting a form of meta-verification by re-verifying their evaluation content, (3) exploring whether there are edge cases that have yet to be identified, and (4) performing backtracking by reviewing the initial problem conditions, the content of the response, and its own evaluation process. We speculate that the ability to apply strategies used for problem-solving to the evaluation process is the key reason why reasoning models show quantitatively good evaluation performance.

**When is process evaluation effective?** The confusion matrix in the bottom row of Figure 4 illustrates the strengths and weaknesses of outcome evaluation and process evaluation. Across different model sizes, we observe that reasoning process evaluators achieve higher precision but lower recall compared

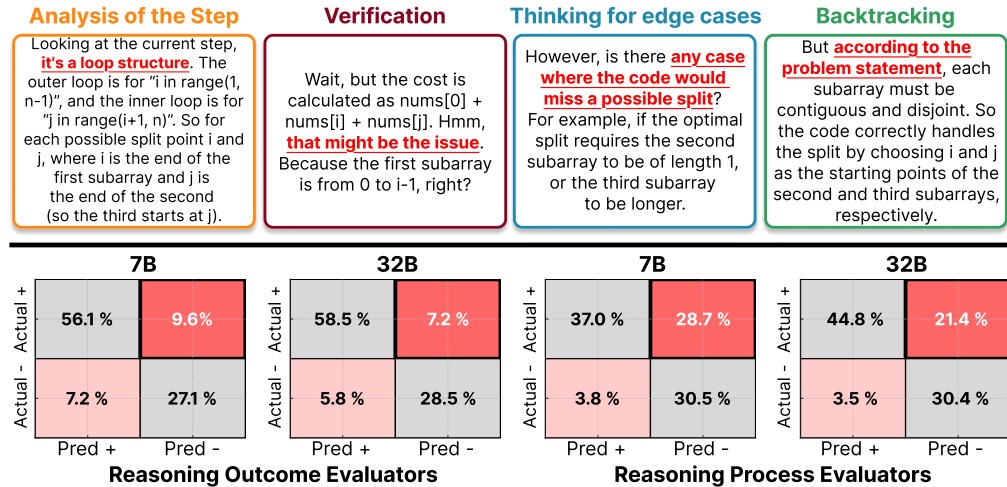

Figure 4: **(Top)** A qualitative example of reasoning process evaluator-generated CoT, which includes diverse reasoning patterns that contribute to improved evaluation performance. **(Bottom)** Reasoning process evaluators judge correct responses as incorrect more frequently than outcome evaluators.

to reasoning outcome evaluators. Consequently, when a reasoning process evaluator predicts that all steps are correct, the final answer is likely to be correct (false positive rates are 3.8% and 3.5%).

## 6 POINTERS TO EXPERIMENTS AND ANALYSES IN APPENDIX

Due to space constraints, we include additional explanations, experiments, and analyses in the Appendix section and provide explicit pointers to relevant content for convenience of reviewers.

- **What is the main difference between prior works on test-time scaling and the main focus of this work?** → In Appendix D, we include a brief overview of prior works on test-time scaling. Also, we summarize the key differences between our work and prior works in Table 7.

- **Why didn't you include generative reward models in your experiments?** → Experimental results of Cloud-RM (Ankner et al., 2024) and Prometheus 2 (Kim et al., 2024c) in our Best-of-$N$ setting is at Appendix C (Table 2) which is an extended section of section 4. We do not include results of Mahan et al. (2024) and Zhang et al. (2024) as they did not release their checkpoints.

- **Does reasoning process evaluators work outside of math and code as well?** → In subsection C.3, we include experimental results on law, engineering, and economics subsets from MMLU-Pro (Wang et al., 2024b) in our Best-of-$N$ setting, which is an extended section of section 4.

- **Does your approach work well when a reasoning model is a generator?** → In Appendix G, we compare ORMs, PRMs, and reasoning models to evaluate responses from a reasoning model.

- **What is a formal explanation for the effectiveness of reasoning process evaluators?** → In Appendix F, we explain how a better evaluator (observed in section 3) can alleviate reward model overoptimization, and translate into enhanced performance of the generator (shown in section 4).

- **Isn't your method more expensive?** → In Appendix E, we explain how we constrain the compute cost of the generator and evaluator. Also, in Appendix I, we further propose a more efficient variant of using a reasoning model as a evaluator while maintaining its superior performance.

## 7 CONCLUSION

We find that applying evaluation-time scaling can improve not only evaluator performance but also the problem-solving capabilities of generators through Best-of-$N$ sampling. We demonstrate that prompting reasoning models to act as outcome or process evaluators are effective methods for scaling evaluation-time compute. Additionally, we show that reasoning process evaluation tends to make more conservative predictions than reasoning outcome evaluation, and that combining reasoning process and reasoning outcome evaluation can result in further performance gains.

## 8 REPRODUCIBILITY STATEMENT

We are committed to ensure the reproducibility of our results. We plan to release a link to the code and artifacts (*e.g.*, model generations) after the anonymity period.

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

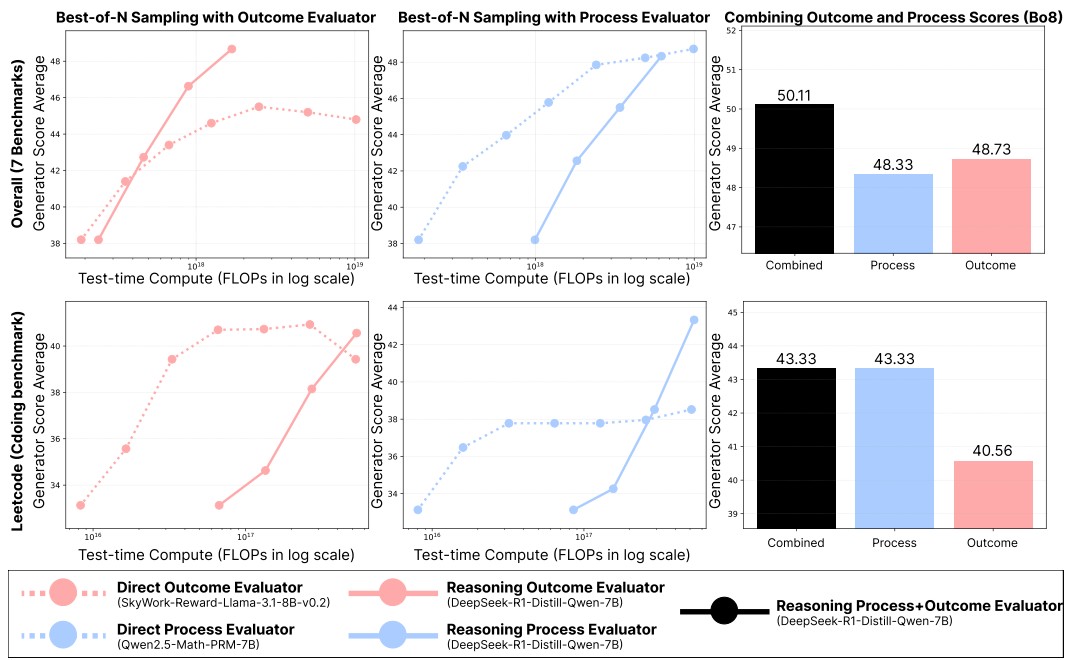

Figure 5: Best-of-$N$ experimental results for $\sim$ 7B scale evaluators. **(left, middle)** We compare direct evaluators (using Best-of-64) against reasoning evaluators (using Best-of-8), with each dot representing a doubling of the number of responses. We find that reasoning evaluators achieve better performance compared to their direct counterparts given a fixed test-time compute budget: see Appendix E for details of our FLOPs calculations. **(right)** Combining outcome and process scores yields further gains.

# A  EXTENDED: EXPERIMENTAL SETTINGS

## A.1  MODEL LIST FOR PROCESSBENCH EXPERIMENTS (SECTION 3)

We examine three varieties of models in our ProcessBench experiments:

- **Direct PRMs:** We experiment with 10 different direct PRMs, representing the state-of-the-art on ProcessBench, from families including `math-shepherd-mistral` (Wang et al., 2024a), `Skywork` (o1 Team, 2024), `RLHFlow` (Xiong et al., 2024), `EurusPRM` (Cui et al., 2025), and `Qwen2.5-Math-PRM` (Zhang et al., 2025).

- **Instruction-tuned Models:** These are models that have been trained using supervised fine tuning and/or RLHF, but have not been explicitly trained for reasoning. We experiment with models from the `Llama-3.1` (Llama Team, 2024), `Llama-3.3` (Llama Team, 2024), and `Qwen2.5` (Team, 2024a) families. We also experiment with `GPT-4-0806` as an outcome evaluator.

- **Reasoning Models:** These are models that have been explicitly trained to perform reasoning using RL, or distilled from models trained to perform reasoning. We examine models from the `DeepSeek-R1-Distill-Qwen` (Guo et al., 2025) and `QwQ` (Team, 2024b) families. We also experiment with `o1-mini` as a reasoning evaluator.

We report results from a subset of these models in the main text: see Table 1. We include results from all listed models in Table 4.

Note that we do not experiment with fine-tuned generative evaluators such as Prometheus (Kim et al., 2023b; 2024a), CLoud-RM (Ankner et al., 2024) and GenRM (Zhang et al., 2024) as these are trained outcome evaluators that cannot be readily employed to detect process errors, as is required by ProcessBench.

Table 2: Full results for Best-of-8 (reasoning evaluators) and Best-of-64 (all other evaluators) experiments using direct outcome evaluators, direct process evaluators, non-reasoning generative evaluators, reasoning outcome evaluators , single-step reasoning process evaluators , reasoning process evaluators , and reasoning outcome + process evaluators . Results were obtained by averaging scores across all 7 evaluation benchmarks and 3 generators as described in section 4.

| Model | $N=1$ | $N=2$ | $N=4$ | $N=8$ | $N=16$ | $N=32$ | $N=64$ |
|---|---|---|---|---|---|---|---|
| **Direct Outcome Evaluators (ORMs)** | | | | | | | |
| Skywork-Reward-Llama-3.1-8B-v0.2 | 38.2 | 41.4 | 43.4 | 44.6 | 45.5 | 45.2 | 44.8 |
| Skywork-Reward-Gemma-2-27B-v0.2 | 38.2 | 41.8 | 43.4 | 44.8 | 45.5 | 45.6 | 45.4 |
| **Direct Process Evaluators (PRMs)** | | | | | | | |
| math-shepherd-mistral-7b-prm | 38.2 | 41.6 | 42.7 | 43.1 | 43.7 | 43.5 | 43.2 |
| Skywork-o1-Open-PRM-Qwen-2.5-1.5B | 38.2 | 42.8 | 44.2 | 45.9 | 46.5 | 46.5 | 46.7 |
| Skywork-o1-Open-PRM-Qwen-2.5-7B | 38.2 | 42.9 | 45.3 | 47.5 | 48.4 | 48.7 | 49.9 |
| RLHFlow/Llama3.1-8B-PRM-Mistral | 38.2 | 40.2 | 40.0 | 39.4 | 38.4 | 37.2 | 35.5 |
| RLHFlow/Llama3.1-8B-PRM-Deepseek | 38.2 | 40.5 | 40.4 | 40.4 | 40.0 | 38.8 | 37.8 |
| Qwen2.5-Math-7B-PRM800K | 38.2 | 41.6 | 43.4 | 45.1 | 45.2 | 45.0 | 44.6 |
| Qwen2.5-Math-PRM-7B | 38.2 | 42.3 | 44.0 | 45.8 | 47.8 | 48.2 | 48.7 |
| Qwen2.5-Math-PRM-72B | 38.2 | 42.9 | 45.4 | 48.2 | 48.9 | 49.8 | 50.6 |
| **Non-Reasoning Generative Evaluators** | | | | | | | |
| Llama3-8B-CLoud-RM | 38.2 | 41.8 | 42.7 | 43.8 | 43.7 | 43.1 | 42.5 |
| prometheus-7b-v2.0 | 38.2 | 40.4 | 41.3 | 41.7 | 41.5 | 40.7 | 39.7 |
| prometheus-8x7b-v2.0 | 38.2 | 40.3 | 40.8 | 41.9 | 41.6 | 41.2 | 40.8 |
| Qwen2.5-72B-Instruct | 38.2 | 42.3 | 44.1 | 45.9 | 45.2 | 45.8 | 45.6 |
| **Reasoning Outcome Evaluators** | | | | | | | |
| DeepSeek-R1-Distill-Qwen-7B | 38.2 | 42.7 | 46.6 | 48.7 | - | - | - |
| DeepSeek-R1-Distill-Qwen-32B | 38.2 | 43.9 | 47.7 | 51.1 | - | - | - |
| **Single-step Reasoning Process Evaluators** | | | | | | | |
| DeepSeek-R1-Distill-Qwen-7B | 38.2 | 41.3 | 44.2 | 45.6 | - | - | - |
| DeepSeek-R1-Distill-Qwen-32B | 38.2 | 41.7 | 44.9 | 48.1 | - | - | - |
| **Reasoning Process Evaluators** | | | | | | | |
| DeepSeek-R1-Distill-Qwen-7B | 38.2 | 42.6 | 45.5 | 48.3 | - | - | - |
| DeepSeek-R1-Distill-Qwen-32B | 38.2 | 43.6 | 46.9 | 50.3 | - | - | - |
| **Reasoning Process + Outcome Evaluators** | | | | | | | |
| DeepSeek-R1-Distill-Qwen-7B | 38.2 | 43.3 | 46.6 | 50.1 | - | - | - |
| DeepSeek-R1-Distill-Qwen-32B | 38.2 | **44.4** | **48.5** | **52.0** | - | - | - |

Table 3: Ablation results of applying different splitting and aggregation functions to direct process evaluators.

| Model | $N=1$ | $N=2$ | $N=4$ | $N=8$ |
|---|---|---|---|---|
| **Splitting**: Heuristic-based, **Aggregation**: min | | | | |
| Qwen2.5-Math-PRM-7B | 38.2 | 42.3 | 44.0 | 45.8 |
| Qwen2.5-Math-PRM-72B | 38.2 | 42.9 | 45.4 | 48.2 |
| **Splitting**: Heuristic-based, **Aggregation**: mean_logits | | | | |
| Qwen2.5-Math-PRM-7B | 38.2 | 40.7 | 41.4 | 41.7 |
| Qwen2.5-Math-PRM-72B | 38.2 | 41.9 | 43.3 | 43.8 |
| **Splitting**: Model-based, **Aggregation**: min | | | | |
| Qwen2.5-Math-PRM-7B | 38.2 | 42.6 | 45.00 | 47.1 |
| Qwen2.5-Math-PRM-72B | 38.2 | 43.2 | 45.8 | 49.9 |

## A.2 MODEL LIST FOR BEST-OF-$N$ EXPERIMENTS (SECTION 4)

- **Direct outcome evaluators**: We experiment with `Skywork-Reward-Llama-3.1-8B-v0.2` (Liu et al., 2024) and `Skywork-Reward-Gemma-2-27B-v0.2` (Liu et al., 2024), which are the state-of-the-art direct outcome evaluators on RewardBench (Lambert et al., 2024) (a widely used benchmark to assess ORMs).

- **Direct process evaluators**: We experiment with `math-shepherd-mistral` (Wang et al., 2024a), `Skywork` (o1 Team, 2024), `RLHFlow` (Dong et al., 2024), `EurusPRM` (Cui et al., 2025), and `Qwen2.5-Math-PRM` (Zhang et al., 2025).

- **Fine-tuned generative evaluators**: While, like reasoning models, these models also produce CoT they, unlike reasoning models, produce short CoT that lack complex reasoning patterns (e.g. self-verification, self-correction, backtracking) as they are not trained to reason using RL. We experiment with `Llama3-8B-CLoud-RM` (Ankner et al., 2024) and the `Prometheus 2` family (Kim et al., 2024c), which are generative models trained specifically for outcome evaluation, as well as `Qwen2.5-72B-Instruct` (Team, 2024a), which we prompt to act as a generative evaluator. We use these models as reasoning outcome evaluators.

- **Reasoning outcome evaluators**: We prompt reasoning models `DeepSeek-R1-Distill-Qwen-7B` and `DeepSeek-R1-Distill-Qwen-32B` to act as reasoning outcome evaluators.

- **Reasoning process evaluator**: We prompt reasoning models `DeepSeek-R1-Distill-Qwen-7B` and `DeepSeek-R1-Distill-Qwen-32B` to act as reasoning process evaluators.

- **Reasoning process + outcome evaluator**: We experiment with the reasoning models `DeepSeek-R1-Distill-Qwen-7B` and `DeepSeek-R1-Distill-Qwen-32B`, using identical models for both the outcome and process evaluation components in each case.

## B  ADDITIONAL RESULTS FOR PROCESSBENCH EXPERIMENTS (SECTION 3)

We include the full experimental results for our ProcessBench experiments (section 3) in Table 4.

## C  ADDITIONAL RESULTS AND DISCUSSION FOR BEST-OF-$N$ EXPERIMENTS (SECTION 4)

We include the full experimental results for our Best-of-$N$ experiments (section 4) in Table 2.

### C.1  FINDINGS FROM EXPERIMENTS

**Evaluation-time Scaling is effective with smaller-sized evaluators as well.**     Similar to Figure 3 with larger-sized evaluators, Figure 5 shows the results of employing smaller-sized evaluators in the Best-of-$N$ setting. The findings from section 4 maintain the same: (1) reasoning evaluators (Best-of-8) outperform or match their direct evaluator counter parts (Best-of-64) while using less amount of compute, (2) reasoning process + outcome evaluation can boost performance, and (3) reasoning evaluators are especially effective for coding.

**Multi-step process evaluation outperforms single-step process evaluation.**     Next, as shown in Table 2, we compare reasoning process evaluators with single-step reasoning process evaluators (see subsection 2.2 for a detailed explanation of the difference). Results show that even when employing the same LM as the evaluator, evaluating each step individually is superior to evaluating all the steps at once, supporting the strength of our approach and effectiveness of evaluation-time scaling.

**Reasoning outcome evaluators outperform specially-trained outcome evaluators.**.     Then, as shown in Table 2, we compare the effectiveness of employing reasoning models as outcome evaluators over using specially-trained outcome evaluators such as CLoud-RM (Ankner et al., 2024) and Prometheus 2 (Kim et al., 2024a). Results show that reasoning models are very effective in our Best-of-$N$ setting. This is notable because it hints that employing LMs with stronger problem-solving capabilities as evaluators is more important than inducing evaluation capabilities through training. Future work could explore recipes for training reasoning models as evaluators.

**Model-based splitting is effective for direct process evaluators as well.**     Lastly, we ablate the effect of applying model-based splitting and the `mean_logits` aggregation function to direct process evaluators. Note that model-based splitting requires the usage of an LM ($M_{split}$) to segment the response into steps, it requires additional compute. Results in Table 3 show that (1) applying a model-based splitting approach is effective and (2) using the mean_logits aggregation function is not.

Table 4: Full experimental results for our ProcessBench experiments (section 3).

| Model | GSM8K | MATH | Olym. Bench | Omni-MATH | Avg. F1 | Δ |
|---|---|---|---|---|---|---|
| **Direct Process Evaluator (PRMs)** | | | | | | |
| math-shepherd-mistral-7b-prm | 47.9 | 29.5 | 24.8 | 23.8 | 31.5 | - |
| Skywork-o1-Open-PRM-Qwen-2.5-1.5B | 57.9 | 48.0 | 16.5 | 18.9 | 35.3 | - |
| Skywork-o1-Open-PRM-Qwen-2.5-7B | 70.8 | 53.6 | 22.9 | 21.0 | 42.1 | - |
| RLHFlow/Llama3.1-8B-PRM-Mistral | 50.4 | 33.4 | 13.8 | 15.8 | 28.4 | - |
| RLHFlow/Llama3.1-8B-PRM-Deepseek | 38.8 | 33.8 | 16.9 | 16.9 | 26.6 | - |
| EurusPRM-Stage1 (7B) | 44.3 | 35.6 | 21.7 | 23.1 | 31.2 | - |
| EurusPRM-Stage2 (7B) | 47.3 | 35.7 | 21.2 | 20.9 | 31.3 | - |
| Qwen2.5-Math-7B-PRM800K | 68.4 | 62.5 | 50.4 | 43.6 | 56.2 | - |
| Qwen2.5-Math-PRM-7B | 82.4 | 77.6 | 67.5 | 66.3 | 73.5 | - |
| Qwen2.5-Math-PRM-72B | **87.3** | **80.6** | **74.3** | **71.1** | **78.3** | - |
| **Single-step Reasoning Process Evaluator** | | | | | | |
| **Instruction-tuned Models** | | | | | | |
| Llama-3.1-8B-Instruct | 24.0 | 15.5 | 9.7 | 10.1 | 14.8 | 0.0 |
| Llama-3.3-70B-Instruct[†] | **82.9** | 59.4 | 46.7 | 43.0 | 58.0 | - |
| Qwen2.5-32B-Instruct | 63.8 | 47.5 | 35.9 | 32.7 | 45.0 | 0.0 |
| Qwen2.5-Math-72B-Instruct[†] | 65.8 | 52.1 | 32.5 | 31.7 | 45.5 | - |
| Qwen2.5-72B-Instruct[†] | 76.2 | **61.8** | **54.6** | **52.2** | **61.2** | - |
| **Reasoning Models** | | | | | | |
| DeepSeek-R1-Distill-Qwen-7B | 68.3 | 61.1 | 48.3 | 40.4 | 54.5 | 0.0 |
| DeepSeek-R1-Distill-Qwen-32B | **83.9** | **78.1** | **72.4** | 67.7 | **75.5** | 0.0 |
| QwQ-32B-Preview | 77.5 | 58.9 | 31.2 | 35.8 | 50.9 | 0.0 |
| QwQ-32B | 79.5 | 77.5 | 71.5 | **69.4** | 74.5 | 0.0 |
| **Reasoning Models** (Self-Consistency) | | | | | | |
| DeepSeek-R1-Distill-Qwen-7B | 69.3 | 67.9 | 54.8 | 51.5 | 60.9 | +6.4 |
| DeepSeek-R1-Distill-Qwen-32B | 82.2 | **80.4** | **76.2** | 72.5 | **77.8** | +2.3 |
| QwQ-32B-Preview | **88.0** | 78.7 | 57.8 | 61.3 | 71.5 | +20.6 |
| QwQ-32B | 81.0 | 78.8 | 74.4 | **72.8** | 76.8 | +2.3 |
| GPT-4-0806[†] | 79.2 | 63.6 | 51.4 | 53.5 | 61.9 | - |
| o1-mini[†] | **93.2** | **88.9** | **87.2** | **82.4** | **87.9** | - |
| **Reasoning Process Evaluator (Ours)** | | | | | | |
| **Instruction-tuned Models** (Multi-step Process Evaluation) | | | | | | |
| Llama-3.1-8B-Instruct | 35.2 | 22.8 | 12.6 | 17.9 | 22.1 | +7.3 |
| Qwen2.5-32B-Instruct | **70.1** | **61.7** | **54.2** | **53.9** | **60.0** | +15.0 |
| **Reasoning Models** (Multi-step Process Evaluation) | | | | | | |
| DeepSeek-R1-Distill-Qwen-7B | 75.5 | 67.3 | 59.8 | 56.6 | 64.8 | +10.3 |
| DeepSeek-R1-Distill-Qwen-32B | 80.3 | 82.2 | **77.0** | 75.0 | 78.6 | +3.1 |
| QwQ-32B-Preview | **81.7** | 79.3 | 70.3 | 69.8 | 75.3 | +24.4 |
| QwQ-32B | 81.5 | **83.6** | 76.8 | **75.1** | **79.3** | +4.8 |
| **Reasoning Models** (Multi-step Process Evaluation + Self-Consistency) | | | | | | |
| DeepSeek-R1-Distill-Qwen-7B | 80.1 | 75.1 | 69.3 | 70.4 | 73.7 | +19.2 |
| DeepSeek-R1-Distill-Qwen-32B | 86.6 | 85.4 | 78.9 | **80.3** | **82.8** | +7.3 |
| QwQ-32B-Preview | **86.8** | **85.7** | 79.0 | 78.0 | 82.4 | +31.5 |
| QwQ-32B | 85.0 | 85.6 | **79.4** | 78.0 | 82.0 | +7.5 |

Table 5: Statistical analysis of mean_logit scores for different sample sizes in Best-of-N evaluation

| N | Mean Score | Std Dev | 96% CI |
|---|---|---|---|
| 1 | 38.14 | 1.712 | (34.31, 41.96) |
| 2 | 43.98 | 1.752 | (40.06, 47.90) |
| 4 | 47.90 | 1.783 | (43.92, 51.89) |
| 8 | 51.58 | 1.782 | (47.60, 55.57) |

## C.2 CHECKING STATISTICAL SIGNIFICANCE OF RESULTS

We run the Best-of-8 experiment twice, with a reasoning evaluator (DeepSeek-R1-Distill-Qwen-32B) calculating scores on 8 responses per instance. For N=8, there are 2 scores; for N=4, there are 4 scores

Table 6: **Reasoning process evaluators are effective outside of math and code as well:** Performance across Economics, Engineering, and Law subsets from MMLU-Pro (Wang et al., 2024b) in Best-of-$N$ experiment setting using different evaluators. Our approach (DeepSeek-R1-Distill-Qwen-7B/32B with Process + Outcome evaluation) outperforms other evaluators while using less number of responses from generators.

| Evaluator | # Generator Responses | Llama-3.1-70B-Instruct | | | DeepSeek-R1-Distill-Qwen-7B | | | Total Avg |
|---|---|---|---|---|---|---|---|---|
| | | Economics | Engineering | Law | Economics | Engineering | Law | |
| **Baselines** | | | | | | | | |
| Greedy N=1 | 1 | 74.5 | 47.0 | 45.5 | 69.5 | 53.0 | 17.5 | 51.17 |
| Skywork-8B (ORM) | 64 | 78.0 | 47.0 | 50.0 | 76.5 | 55.5 | 23.5 | 55.08 |
| Skywork-27B (ORM) | 64 | 80.5 | 42.5 | 51.0 | 74.5 | **58.0** | 23.5 | 54.92 |
| Qwen-PRM-7B | 64 | 76.0 | 50.5 | 47.5 | 74.0 | 55.0 | 21.5 | 54.08 |
| Qwen-PRM-72B | 64 | 74.5 | 41.5 | **48.0** | 78.0 | 52.0 | 26.0 | 53.33 |
| **Ours** | | | | | | | | |
| DeepSeek-R1-Distill-Qwen-7B (Process + Outcome) | 8 | 79.0 | 51.5 | 47.0 | 75.0 | 52.0 | 28.5 | 55.33 |
| DeepSeek-R1-Distill-Qwen-32B (Process + Outcome) | 8 | **81.0** | **56.5** | 47.5 | **79.0** | 54.0 | **32.0** | **58.33** |

(index 0∼3 from first run, index 4∼7 from first run, index 0∼3 from second run, index 4∼7 from second run); and so on. As shown in Table 5, all scores across N=1,2,4,8 are statistically significant at the 96% confidence level.

### C.3 TESTING ON DOMAINS OUTSIDE OF MATH AND CODE

**Experimental setting**    To evaluate whether our reasoning process evaluator extends beyond math and coding tasks, we conduct experiments on 500 samples from each of the economics, engineering, and law subsets of MMLU-Pro (Wang et al., 2024b). To sample responses, we use both a reasoning model (DeepSeek-R1-Distill-Qwen-7B) and a non-reasoning model (Llama-3.1-70B-Instruct). For each question, we sample 64 candidate responses from both generators.

**Experimental results**    The results are shown in Table 6. While baseline evaluators (ORMs and PRMs) operate on all 64 candidates, our reasoning-based evaluator processes only 8 responses. Despite this smaller candidate set, the results show that our 7B evaluator outperforms both the 8B ORM and the 7B PRM, and our 32B evaluator outperforms both the 27B ORM and the 32B PRM. These findings, consistent with the trends observed in Figure 3 and Figure 5, demonstrate that evaluation-time scaling with reasoning evaluators remains effective across diverse domains, including economics, engineering, and law, thereby confirming that our methodology generalizes beyond math and coding.

## D    RELATED WORK

### D.1    SCALING TEST-TIME COMPUTE

Increasing compute by enlarging model size or expanding training data has long been one of the key methods to improve LM performance during training time (Kaplan et al., 2020; Hoffmann et al., 2022; Longpre et al., 2023; Kim et al., 2023a). However, as it becomes increasingly difficult to obtain high-quality data sufficient for steady advancement of LM performance, a new paradigm has gained attention: scaling compute at test-time instead of training-time (Villalobos et al., 2024; Welleck et al., 2024; Wu et al., 2024b). This approach is attracting interest as a method that can enhance LM performance in a way that complements training time compute. The main approaches to scaling up test-time compute include, first, leveraging sufficient compute at test-time by training reasoning models that generate longer and qualitatively different Chain-of-Thought (CoT) compared to existing chat models (Guo et al., 2025; Yeo et al., 2025; Muennighoff et al., 2025), and second, using inference-time algorithms such as Best-of-$N$ sampling at test-time (Sun et al., 2024; Welleck et al., 2024). Existing works on test-time compute have primarily focused on improving LM's problem-solving capability, whereas we focus on scaling compute for evaluation to enhance evaluators' capabilities by assessing each response step with process evaluation and generating long CoT for precise evaluation.

Table 7 summarizes the difference between representation works on test-time scaling (on the generator side) versus our approach on scaling evaluation-time compute.

Table 7: **Test-time scaling on generator vs. evaluator:** Summary of methods and trade-offs across prior work and ours. Unlike prior work that mainly scales the generator while fixing the evaluator, we are the first to explicitly study scaling *evaluator* test-time compute. We show that scaling evaluation-time strategies can directly improve evaluator performance, opening new directions for test-time evaluation research. Generator-side variants include (A) multiple responses (parallel), (B) beam/lookahead with a PRM (parallel), (C) sequential revision. Evaluator-side variants include (X) a PRM, (Y) self-consistency on the evaluator (parallel), and (Z) a reasoning model as a process evaluator (sequential).

| Paper | Method for Scaling Test-time Compute on Generator side | Method for Scaling Test-time Compute on Evaluator side | Investigated Trade-off |
|---|---|---|---|
| **Snell et al. (2024)** | **(A)** Generating Multiple Responses (Parallel), **(B)** Beam/Lookahead search using a PRM (Parallel), **(C)** Sequential Revision (Sequential) | **(X)** Using a PRM | **"A+X" vs "B+X"**: Test-time scaling for search with verifiers; **"A+X" vs "C+X"**: Test-time scaling with revisions. |
| **Muennighoff et al. (2025)** | **(A)** Generating Multiple Responses (Parallel), **(B)** Budget Forcing (Sequential) | **(X)** Using a PRM | **"A+X" vs "B+X"**: Parallel vs sequential scaling. |
| **Brown et al. (2024)** | **(A)** Generating Multiple Responses | **(X)** Using an ORM | **"A+X"**: Scaling laws as the number of responses increases. |
| **Ours** | **(A)** Generating Multiple Responses (Parallel) | **(X)** Using an ORM or PRM; **(Y)** Applying self-consistency to the evaluator (Parallel); **(Z)** Utilizing a reasoning model as a process evaluator (Sequential) | **"X" vs "Y" vs "Z"**: Evaluation-time scaling (Sec. 3); **"A+X" vs "A+Z"**: Translating improved reasoning to problem solving (Sec. 4). |

## D.2 LANGUAGE MODEL EVALUATORS

Accurately verifying the outputs generated by a language model (LM) is crucial for understanding the types of errors it frequently makes and identifying its limitations (Liang et al., 2023; Mondorf and Plank, 2024; Zheng et al., 2024a; Lee and Hockenmaier, 2025). Recently, evaluators—LMs that assess the quality of a given response (also referred to as verifiers, reward models, or judges in the literature)—have gained significant attention for their ability to provide precise assessments of LM outputs (Zheng et al., 2024b; Ye et al., 2024; Kim et al., 2023b; Lambert et al., 2024; Gu et al., 2024). Evaluators are not only used for benchmarking purposes but also for enhancing the LM's problem solving capabilities (Cobbe et al., 2021; Uesato et al., 2022; Lightman et al., 2024; Wang et al., 2024a; Sun et al., 2024; Wu et al., 2024a).

When an evaluator fails to assess accurately, it may result in unintended consequences for the purpose it is serving (Gao et al., 2023; Coste et al., 2024; Moskovitz et al., 2024). For example, if an evaluator fails to provide accurate judgments, even if a specific LM being evaluated performs well, its true capabilities may be misrepresented due to the errors stemming from the evaluator's limitations (Dubois et al., 2024; Li et al., 2024; Kim et al., 2024b). Also, when integrating an evaluator into an inference-time algorithm, the imperfection of the evaluator might result in diminishing returns even when using more test-time compute (Gao et al., 2023; Rafailov et al., 2024). These limitations highlight the need for more robust evaluators that can generalize in diverse contexts. While Kalra and Tang (2025) has examined debate-based strategies and usage of larger models as evaluators to scale up evaluation-time compute, our work specifically focuses on 'using reasoning models as process evaluators' to demonstrate the effectiveness of evaluation-time scaling.

## D.3 FUTURE DIRECTIONS

Looking ahead, we envision our research enabling advances in two areas. First, evaluation-time scaling can provide better training signals. In particular, it is widely known that generators often develop undesirable traits through reward model over-optimization when given imprecise rewards during reinforcement learning (Stiennon et al., 2020; Bai et al., 2022; Ouyang et al., 2022; Huang et al.,

2022); investigating whether reasoning process evaluators can mitigate this represents a promising direction. Second, future work could explore whether reasoning evaluators can be improved through training. Existing trained evaluators do not leverage the long CoTs that have proven effective in this work, yet we believe that training such models may be key to further enhancing LM evaluation capabilities.

## E    APPROXIMATION OF TEST-TIME COMPUTE

For approximating inference-time compute as in Figure 3 and Figure 5, we follow Snell et al. (2024) and Son et al. (2025). Specifically, inference compute cost can be asymptotically approximated by:

$$C \in O(N \times L), \tag{9}$$

where $C$ is the computation cost, $N$ is the number of parameters and $L$ is the number of tokens. Therefore, we use $N \times L$ as a relative inference compute for a single inference call.

For instance, consider a Best-of-8 case where the generator of size 70B generates total 1,000 tokens in average (**generation-time compute** for response), and the reasoning outcome evaluator of size 7B generates total 3,000 tokens in average (**evaluation-time compute** for CoT and judgment). In this case, the approximate inference-time compute can be calculated as:

$$8 \times ((70 \times 10^9 \times 1000) + (7 \times 10^9 \times 3000)) = 7.28 \times 10^{17}$$

On a high level, when we break down inference-time compute into generator-time compute and evaluation-time compute, $70 \times 10^9 \times 1000$ corresponds to the generation-time compute and $7 \times 10^9 \times 3000$ corresponds to the evaluation-time compute. Therefore, Best-of-8 with reasoning process evaluators (that spends more evaluation-time compute than generation-time compute) requires similar inference-time compute compared to Best-of-64 with direct evaluators (that spends more generation-time compute than evaluation-time compute).

## F    FORMAL EXPLANATION OF WHY EVALUATION-TIME SCALING CAN IMPROVE THE GENERATOR'S PERFORMANCE

We provide a more formal explanation for why evaluation-time scaling can be more effective than the conventional approach of sampling a larger number of responses with a weaker evaluator. The key intuition is that a stronger evaluator mitigates the phenomenon of reward model over-optimization, where imperfect evaluators overvalue certain responses due to noise or bias.

Let $u(x)$ denote the oracle quality of a candidate response $x$. An evaluator provides a surrogate score

$$E(x) = u(x) + \delta(x), \tag{10}$$

where $\delta(x)$ captures evaluation error (biases or noise). In Best-of-$N$, given $N$ candidates $Y = \{y_1, \ldots, y_N\}$, the oracle-best candidate is

$$y_N^* = \arg\max_{y_i \in Y} u(y_i), \tag{11}$$

while the candidate chosen by the evaluator is

$$\arg\max_{y_i \in Y} \big(u(y_i) + \delta(y_i)\big). \tag{12}$$

Because the evaluator is imperfect, the selected response may be suboptimal if it receives an erroneously high $\delta(y_i)$. The probability of such mis-selection grows with $N$, since larger candidate sets increase the chance of some $y_i$ having a large positive error. This effect, often referred to as *reward model over-optimization* or *reward hacking*, undermines the benefits of scaling the generator alone.

Suppose we instead use an improved evaluator

$$E'(x) = u(x) + \delta'(x), \tag{13}$$

where $\delta'(x)$ has lower variance or higher fidelity with respect to $u(x)$. In this case, the probability of correctly selecting the oracle-best candidate improves:

$$P\Big(\arg\max_i E'(y_i) = y_N^*\Big) \; > \; P\Big(\arg\max_i E(y_i) = y_N^*\Big). \tag{14}$$

Crucially, our results show that even with fewer candidates ($n \ll N$), scaling evaluation-time compute with a stronger evaluator can yield

$$P\Big(\arg\max_i E'(y_i) = y_n^*\Big) \; > \; P\Big(\arg\max_i E(y_i) = y_N^*\Big). \tag{15}$$

Thus, despite reducing the number of samples, improved evaluation quality can more than compensate for the loss in oracle score.

This explanation aligns with our experimental results provided in section 4. First, in the low-budget regime (e.g., 1–2 candidates), investing in evaluation compute provides little benefit, since oracle quality is too low for the evaluator to meaningfully distinguish among responses. In this regime, additional sampling dominates. Second, as the number of candidates grows (16–64), a weaker evaluator (e.g., PRM) becomes increasingly vulnerable to mis-selection, while our approach continues to benefit from reduced error. Finally, scaling evaluation-time compute produces smooth gains in general: when moving from 1 to 8 responses, both the oracle quality improves and the stronger evaluator reliably identifies the best candidate, leading to consistent upward scaling curves.

Table 8: **Reasoning models can self-evaluate its response effectively when functioning as a reasoning process evaluator:** We prompt DeepSeek-R1-Distill-Qwen-7B to act as both a generator and its own Best-of-$N$ reasoning process + outcome evaluator (**self-evaluation**). We also report the improvements when using Eurus-2-SFT, Llama3.1-70B-Instruct, and Qwen2.5-7B-Instruct as a generator and DeepSeek-R1-Distill-Qwen-7B as an evaluator for relative comparison. We measure performance improvements (between Best-of-1 and Best-of-8) as a percentage of the gap between Best-of-1 and oracle performance, denoted as **Gap Recovered**. We find that the gains associated with this are comparable to or larger than the gains associated with Best-of-$N$ using the same evaluation strategy on the outputs of instruction-tuned generators.

| Generator | $N = 1$ | $N = 8$ | **Oracle** | **Gap Recovered (%)** |
|---|---|---|---|---|
| **AIME24** | | | | |
| Eurus-2-SFT | 13.3 | 20.0 | 20.0 | **100.0** |
| Llama3.1-70B-Instruct | 16.7 | 23.3 | 36.7 | 33.0 |
| Qwen2.5-7B-Instruct | 10.0 | 16.7 | 23.3 | 50.4 |
| Self-Evaluation (Outcome Eval on CoT + Process Eval on summary) | 50.0 | 73.3 | 83.3 | 70.0 |
| Self-Evaluation (Outcome Eval on summary + Process Eval on summary) | 50.0 | 66.7 | 83.3 | 50.2 |
| **AMC23** | | | | |
| Eurus-2-SFT | 31.1 | 45.3 | 62.7 | 44.9 |
| Llama3.1-70B-Instruct | 26.8 | 45.3 | 65.1 | 48.3 |
| Qwen2.5-7B-Instruct | 36.6 | 51.6 | 69.9 | 45.0 |
| Self-Evaluation (Outcome Eval on CoT + Process Eval on summary) | 85.5 | 88.0 | 92.8 | 34.2 |
| Self-Evaluation (Outcome Eval on summary + Process Eval on summary) | 85.5 | 89.2 | 92.8 | **50.7** |

Table 9: **Reasoning models can self-evaluate its response more effectively than using direct evaluators:** Comparison of direct evaluators (ORMs and PRMs) with 64 candidate responses versus our reasoning process evaluator with only 8 responses. We generate the responses with DeepSeek-R1-Distill-Qwen-7B and test different evaluators in Best-of-$N$ setting. Our approach achieves the highest accuracy on both benchmarks.

| Evaluator | # Candidate Responses | AIME-2024 | AMC-2023 |
|---|---|---|---|
| Greedy | 1 | 50.0 | 85.54 |
| Skywork-8B (ORM) | 64 | 60.0 | 77.10 |
| Qwen-PRM-7B | 64 | 40.0 | 73.50 |
| Skywork-27B (ORM) | 64 | 60.0 | 88.00 |
| Qwen-PRM-72B | 64 | 56.66 | 85.54 |
| DeepSeek-R1-Distill-Qwen-7B (Ours; Process + Outcome) | 8 | **66.7** | **89.2** |

# G    SELF-EVALUATION OF REASONING MODELS

In section 4, we study whether using reasoning models as Best-of-$N$ evaluators improves the problem-solving capabilities of instruction-tuned models. This naturally raises the question of whether similar gains could be achieved if a reasoning model is used both as the generator and the evaluator (*i.e.*, **self-evaluation**). To explore this, we conduct a preliminary experiment in which DeepSeek-R1-Distill-Qwen-7B is used both to generate responses and to evaluate them. Due to computational constraints, we only assess self-evaluation on AIME24 and AMC23 in the Best-of-8 setting.

**The challenge of assessing long CoTs**    A notable characteristic of current reasoning models is that their CoTs (bookended by "<think></think>" tokens) are often lengthy and include numerous reasoning steps. This presents practical challenges for evaluation, as the evaluator must be able to handle long contexts and accurately assess text that includes exploratory reasoning, backtracking, and self-correction steps. However, rather than evaluating the entire CoT trace, we could instead evaluate the reasoning **summary** that is automatically produced by the reasoning model after the CoT. This reasoning summary condenses the exploratory CoT into a more concise form resembling the CoT of instruction-tuned models: see Appendix G.1 for an illustrative example.

**Experimental setting**    We first generate responses to the AIME24 and AMC23 datasets with DeepSeek-R1-Distill-Qwen-7B, setting $t = 0.6$ and $N = 8$. We then perform Best-of-$N$ by prompting DeepSeek-R1-Distill-Qwen-7B to act as a reasoning outcome + process evaluator, following the method described in subsection 2.3. In addition to reporting Best-of-1 and Best-of-8 performance, we also report the percentage of the performance gap between the Best-of-1 and oracle performances

recovered by Best-of-8 (denoted as **Gap Recovered**). For self-evaluation, we always perform reasoning process evaluation on the output summaries, whereas we experiment with reasoning outcome evaluation on both the summaries and the entire CoT. We document our findings in Table 8.

**Main Results**    Our results in Table 8 provide preliminary evidence that reasoning models can be used to improve their own outputs through Best-of-$N$. Specifically, the gains associated with this (Gap Recovered for DeepSeek-R1-Distill-Qwen-7B) is comparable to or larger than the gains associated with Best-of-$N$ on the outputs of the other generators (Eurus-2-SFT, Llama-3.1-70B-Instruct, and Qwen2.5-7B-Instruct) when using the same evaluator (DeepSeek-R1-Distill-Qwen-7B).

**Evaluating the summary is as effective as evaluating entire CoT**    We also find that performing the outcome evaluation component of our reasoning process + outcome evaluation strategy on thoughts improves over outcome evaluation on summaries for AIME, whereas the opposite is true for AMC, although both strategies achieve notable gains over Best-of-1 on both datasets. We hope that our results encourage further investigation into self-evaluation strategies for reasoning models.

**Comparison with direct evaluators**    We further evaluate the performance of state-of-the-art ORMs and PRMs under the same setting, fixing the generator to DeepSeek-R1-Distill-Qwen-7B and varying only the evaluators. While baseline evaluators operate on 64 candidate responses, our reasoning-based evaluator uses only 8 responses from the generator. The last row of Table 9 corresponds to our proposed method, which follows the "Outcome Eval on summary + Process Eval on summary" configuration reported in Table 8.

The results demonstrate that generating a moderate number of responses ($N = 8$) with a reasoning model and evaluating them with a reasoning evaluator achieves higher accuracy than generating a much larger set of responses ($N = 64$) evaluated with ORMs or PRMs. This finding highlights the complementary role of reasoning: the reasoning model not only improves problem solving through its longer chains of thought, but also enhances evaluation fidelity. Taken together, these results confirm that using reasoning models for both generation and evaluation provides a synergistic effect that surpasses conventional test-time scaling approaches.

## G.1 EXAMPLE OF A REASONING MODEL'S OUTPUT

We provide an example of DeepSeek-R1-Distill-Qwen-7B's response to an input from AMC23. The pink box contains the chain-of-thought portion of the response, while the green box contains the summary. The CoT and the summary are separated by a "`</think>`" token.

---

**Problem.** In the state of Coinland, coins have values $6, 10$, and $15$ cents. Suppose $x$ is the value in cents of the most expensive item in Coinland that cannot be purchased using these coins with exact change. What is the sum of the digits of $x$?

**Response**

> ### Chain-of-thoughts
>
> Okay, so I have this problem about Coinland where the coins are 6, 10, and 15 cents. I need to find the largest amount of money that can't be made using these coins, and then find the sum of the digits of that amount. Hmm, interesting.
>
> First, I remember something about the Frobenius number, which is the largest number that can't be expressed as a combination of given coin denominations. But I think that's mainly for two coins. Since we have three here, it might be more complicated. Maybe I should try to see if there's a pattern or a way to approach it.
>
> Let me list out some numbers and see which ones can be made with 6, 10, and 15. Maybe I can find a point beyond which all numbers can be made, and then the largest one before that would be the answer.
>
> Starting from 1 cent and going upwards:
>
> ...
>
> So, starting from 30, every number can be made. Therefore, 29 is indeed the largest number that cannot be formed.
>
> So, the answer is 29, and the sum of its digits is $2 + 9 = 11$.
>
> **Final Answer**
> The sum of the digits of the largest non-makable amount is $\boxed{11}$.`</think>`

> ### Summary
>
> To determine the largest amount of money that cannot be made using coins of 6, 10, and 15 cents, we start by listing the numbers and checking if they can be formed using these coins.
>
> We find that the numbers 1, 2, 3, 4, 5, 7, 8, 9, 11, 13, 14, 17, 19, 23, and 29 cannot be formed. We then check numbers beyond 29 and find that all numbers from 30 onwards can be formed using combinations of 6, 10, and 15 cents.
>
> Thus, the largest number that cannot be formed is 29. The sum of the digits of 29 is $2 + 9 = 11$.
>
> $\boxed{11}$

# H ADDITIONAL ANALYSES (SECTION 5

## H.1 WHEN DOES PROCESS EVALUATION FAIL?

**When does process evaluation fail?** We manually analyze 90 false negative cases from the 7B reasoning process evaluator on MATH-500, OlympiadBench, and GPQA. We identify two common failure modes: (1) the evaluators incorrectly flag correct steps as errors, and (2) the solutions contain actual reasoning errors despite reaching correct conclusions, a phenomenon known as *unfaithful reasoning* (Lyu et al., 2023; Wang et al., 2025). Our analysis reveals that 44.4% of flagged steps contain genuine errors (see Figure 6), confirming that unfaithful reasoning significantly contributes to the discrepancy between process and outcome evaluation. Further analyses with examples can be found in Table 10.

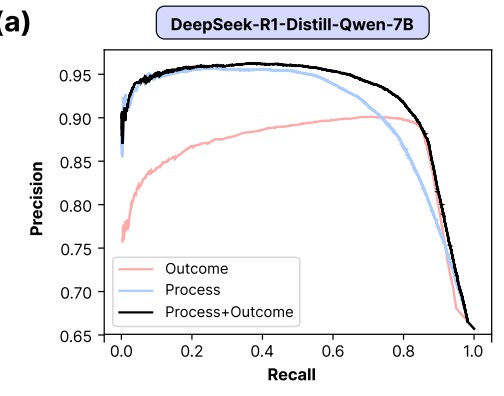

Figure 6: Error pattern analysis of false positive cases from reasoning process evaluators. Please see Appendix H.3 for more details.

## H.2 PRECISION-RECALL CURVE

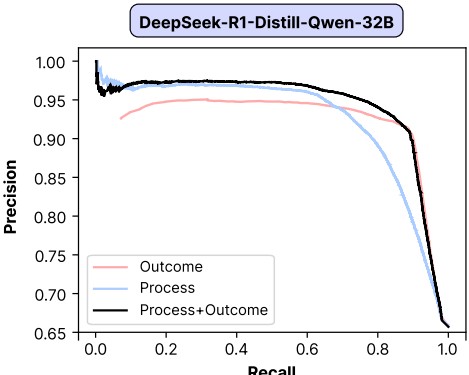

Figure 7: P-R curve of reasoning outcome evaluators and process evaluators. While reasoning process evaluators achieve higher precision in the low-recall region, reasoning outcome process evaluators achieve better performance in the high-recall region. Combining both scores is an optimal strategy that achieves better precision than outcome and process evaluation in any region.

Figure 4 (Bottom) displays the confusion matrix of reasoning outcome evaluators and process evaluators using a constant classification threshold (0.5). However, one can also plot a precision-recall curve (P-R curve) by varying the threshold. In this plot, recall=0 indicates that the threshold is high ($\sim 1.0$) and all correct responses are classified as negative, and recall=1 indicates that the threshold is small ($\sim 0.0$) and all correct responses are classified as positive.

The resulting P-R curve is shown in Figure 7. Aligned to the findings of section 5, the curves show that reasoning process evaluators are more *conservative* classifiers than reasoning outcome evaluators. The curve indicates that if the process evaluator assigned a high score, it is more likely that the final answer is correct (high precision in low-recall region). However, outcome evaluators achieve better overall accuracy (high precision in high-recall region), indicating that process evaluators are more likely to assign relatively low scores to correct responses than outcome evaluators. Combining outcome and process scores results in similar trends to process evaluators in the low-recall region and to outcome evaluators in the high-recall region, achieving the best of both worlds.

## H.3 QUALITATIVE ANALYSIS

In this section, we briefly present the criteria for manual analysis on *steps that lead to a correct answer but are predicted as incorrect by reasoning process evaluators.*

First, we randomly sample 90 responses from MATH500, OlympiadBench, and AMC generated using `Llama-3.1-70B-Instruct`, where the response's final answer is correct but the reasoning process evaluator (DeepSeek-R1-Distill-7B) flag a step-level error. The three datasets were chosen because (1) they cover a diverse range of problems including relatively easier (MATH500), medium-level (AMC), and hardest problems (OlympiadBench)[2], and (2) these three datasets demonstrate the most significant gap between reasoning outcome evaluators and reasoning process evaluators in Best-of-$N$ setting.

The authors manually analyzed the first erroneous step flagged by the reasoning process evaluators. The flagged steps are classified into *errors* and *non-errors*. Errors include clear logical or mathematical errors or unjustified falsifiable statements, whereas non-errors include correct reasoning steps, assumptions, and text unrelated to reasoning. The taxonomy is displayed within Table 10.

| Error Type | Example |
|---|---|
| **Error** (ignored) 35.6% | **Problem.** In the $xy$-plane, a circle of radius 4 with center on the positive $x$-axis is tangent to the $y$-axis at the origin, and a circle with radius 10 with center on the positive $y$-axis is tangent to the $x$-axis at the origin. What is the slope of the line passing through the two points at which these circles intersect? The final answer can be written in the form $\frac{m}{n}$, where $m$ and $n$ are relatively prime positive integers. What is $m + n$? 
 ... 
 **Step 5.** When $y = 0, x = 0$. This is the **origin, which is not the intersection point of the two circles** (*cf. two circles intersect at the origin*), so we discard this solution. When $y = \frac{80}{29}, x = \frac{5y}{2} = \frac{5 \cdot 80}{2 \cdot 29} = \frac{200}{29}$. So, the coordinates of the intersection point are $(\frac{200}{29}, \frac{80}{29})$. |
| **Error** (explicitly corrected) 8.9% | **Problem.** Zach has three bags and a bunch of pencils to be placed into the bags. He is told to place the greatest number of pencils possible into each of the three bags while also keeping the number of pencils in each bag the same. What is the greatest number of pencils he could have left over? 
 ... 
 **Step 2.** Since Zach has to distribute the pencils evenly among the three bags, the total number of pencils must be a multiple of 3. However, **to have the greatest number of pencils left over, we need to consider the scenario where the total number of pencils is 1 more than a multiple of 3.** (*cf. corrected by Step 5*) 
 **Step 5.** However, we are looking for the greatest number of pencils left over. Since 3x + 2 is the largest multiple of 3 plus a remainder, the greatest number of pencils Zach could have left over is 2. |
| **Non-error** (reasoning) 23.3% | **Problem.** What is the value of $(3x - 2)(4x + 1) - (3x - 2)4x + 1$ when $x = 4$? 
 ... 
 **Step 2.** First, substitute x = 4 into the given expression: $(3(4) - 2)(4(4) + 1) - (3(4) - 2)4(4) + 1$ |
| **Non-error** (planning/restatement) 18.9% | **Problem.** There are two increasing sequences of five consecutive integers, each of which have the property that the sum of the squares of the first three integers in the sequence equals the sum of the squares of the last two. Determine these two sequences. 
 ... 
 **Step 1.** To solve this problem, let's break it down step by step. Let the first sequence be: a, a+1, a+2, a+3, a+4 |
| **Non-error** (segmentation) 13.3% | **Problem.** Suppose that $f(a) = 2a^2 - 3a + 1$ for all real numbers $a$ and $g(b) = \log_{1/2} b$ for all $b > 0$. Determine all $\theta$ with $0 \le \theta \le 2\pi$ for which $f(g(\sin \theta)) = 0$. 
 ... 
 **Step 9.** This gives us two possible values for $u$: $u = 1/2$ and $u = 1$. 
 **Step 10.** Now, we substitute back in for $u$. $\log_{1/2}(\sin \theta) = 1/2 \log_{1/2}(\sin \theta) = 1$ 
 (*cf.* $\log_{1/2}(\sin \theta) = 1/2$ \n $\log_{1/2}(\sin \theta) = 1$) |

Table 10: Examples of steps flagged as an error by reasoning process evaluators.

---

[2]Despite that AIME is the hardest dataset with the lowest average scores, the small size (30 problems) makes it insufficient for manual error analysis.

# I  ANALYSIS: HOW DO MIXING PROCESS AND OUTCOME EVALUATION LEAD TO SUPERIOR PERFORMANCE?

Our reasoning process + outcome evaluator baseline averages the scores from reasoning process evaluators and reasoning outcome evaluators and has shown to be effective in our Best-of-$N$ experiments at section 4. To better understand the reasons behind this, we first analyze how the results change when mixing with different ratios ($\alpha$ values). We perform a grid search of the $\alpha$ value from $0.0$ to $1.0$ with step size $0.1$ and find that the optimal $\alpha$ is skewed towards the outcome score, where weighting process score more than 0.5 causes the performance to decline. (Figure 8) [3]

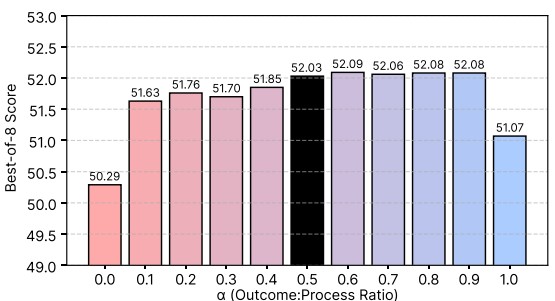

Figure 8: Optimal mixing rate between reasoning outcome scores and process scores is skewed towards outcome evaluator. Increasing the proportion of process scores leads to reduced scores.

The optimal mixing rate highly (but not entirely) skewed towards outcome evaluators suggests that process evaluation serves as a **tie-breaker** for outcome evaluation when merged. Since reasoning outcome evaluators output tokens `0/1` as the correctness label, the scores (token probabilities of the label 1) are indistinguishable between responses labeled as correct or wrong. In process+outcome evaluators, process scores can be applied to break ties in responses by penalizing process errors, leading to improved Best-of-$N$ accuracy.

To prove this intuition that process evaluators can further rerank responses indistinguishable by outcome evaluators, we explore an alternative of $\alpha$-weighted average version of process+outcome evaluators, **2-stage prompting** (Figure 9). In this setting, responses are first filtered using the outcome score. Responses with outcome scores higher than 0.99 were analyzed by process evaluators, selecting the top response. Therefore, responses with *low outcome scores but high process scores* cannot be chosen as the final candidate. Intuitively, the 2-stage prompting's performance is strictly bounded by outcome evaluator's recall and process evaluator's precision, whereas the soft merging of the process+outcome evaluator offers more flexibility.

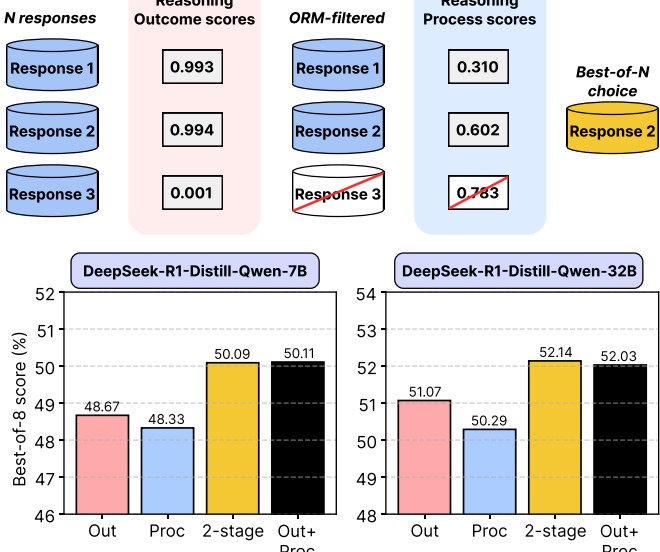

Figure 9: While reasoning outcome evaluators are generally better at finding the correct answer due to high recall, reasoning process evaluators can perform tie-breaking with high accuracy among outcome evaluator-filtered samples, outcome scores, and even process+outcome scores. This suggests that process evaluators can efficiently filter false positives, *i.e.*, the responses that outcome evaluators classified as correct but contains process-level errors.

The results show that the performance of the 2-stage prompting is significantly higher than that of outcome evaluator and is almost identical to that of process+outcome evaluator. As the difference between reasoning outcome scores is extremely small, using only outcome scores might not entirely reflect the quality of the responses and lead to a suboptimal Best-of-$N$ performance. However, process evaluators can further

---

[3]Throughout this section, the reported scores are from DeepSeek-R1-Distill-Qwen-32B.

rerank responses that outcome evaluators assign indistinguishable scores as shown in 2-stage prompting, which is the key aspect of the optimal Best-of-$N$ performance of reasoning process+outcome evaluator.

One benefit of 2-stage prompting is that it reduces the inference cost by only applying the reasoning process evaluation to responses that passed the outcome evaluation. While different heuristics can be applied to optimize the compute while retaining the Best-of-$N$ performance (*e.g.* perform process evaluation only if outcome evaluators classified responses with different final answers as correct), we leave this direction as future work.

## J    ANALYSIS: HOW DOES PROBLEM DIFFICULTY AFFECT OUTCOME AND PROCESS EVALUATION?

Another important factor regarding Best-of-$N$ performance is the *problem difficulty*, often estimated by the fraction of correct answers out of $N$ responses. The fraction value is empirically important because if there are more correct answers, there is a higher chance of selecting a response with a correct answer. However, if there are only a few correct answers, it is generally challenging to rank the correct answer at the top.

As seen in the relative performance (Figure 10), Reasoning outcome evaluators outperform process evaluators in difficult problems, whereas process evaluators achieve higher Best-of-$N$ accuracy in relatively easier problems. This can be explained by the conclusion of Appendix I, that reasoning process evaluators are conservative classifiers and often assigns low score to responses with correct answers. However, if there is a sufficient amount of correct responses, the conservative nature of process evaluators prevents choosing responses with incorrect steps, increasing the expected quality of the top response.

The problem difficulty also affects the performance gap between reasoning process+outcome evaluators and reasoning outcome evaluators. Fewer correct answers increase the chance of *false positives* in outcome evaluators, where they assign high (>0.99) scores to responses with incorrect answers. When using process scores together, such false positives can be effectively reranked and filtered as shown in Appendix I, leading to improved performance in the Best-of-$N$ setting.

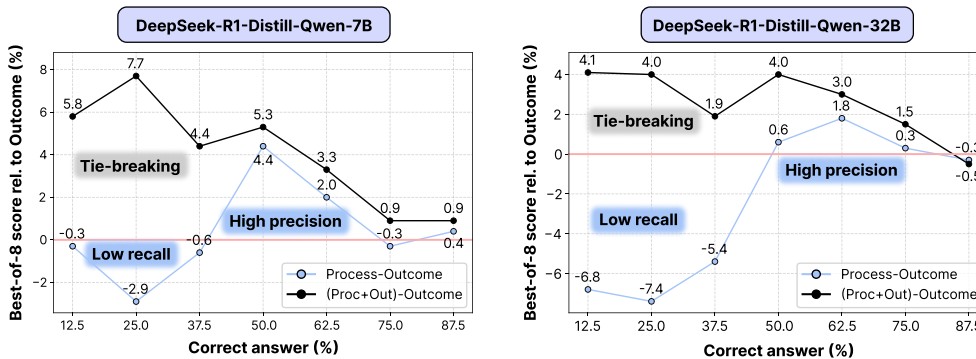

Figure 10: While reasoning process evaluators achieve low Best-of-$N$ score compared to reasoning outcome due to low recall (section 5), Reasoning process+outcome evaluators outperform outcome evaluators by leveraging the tie-breaking ability of process evaluators. Both effects are more significant in difficult problems, where the response generator models are unlikely to find the correct answer.

## K  PROMPTS FOR REASONING EVALUATORS

We include the prompts used to elicit reasoning models as process and outcome evaluators:

---

**Reasoning process evaluator prompt:**

The following is a math problem and a solution (split into paragraphs, enclosed with tags and indexed from 0):

**Problem**
{problem}

**Previous Paragraph(s)**
{previous_paragraphs}

**Current Paragraph**
{current_paragraph}

**Instructions**
Your task is to decide whether the current paragraph is correct or not. If the current paragraph is correct, return the index of 1 and if not, return the index of 0.

Don't try to solve the problem. Your task is only to critique the current paragraph.

Please put your final prediction (i.e., the correctness, which must be 0 or 1) in boxed{{}}. Every output must therefore contain either $\boxed{1}$ or $\boxed{0}$.

You should only consider the logical correctness of the current paragraph, not whether it is useful or has the potential to lead to the correct answer.

---

**Reasoning outcome evaluator prompt:**

The following is a math problem and a solution (split into paragraphs, enclosed with tags and indexed from 0):

**Problem**
{problem}

**Response**
{response}

**Instructions**
Your task is to decide whether the solution is correct or not. If the solution is correct, return the index of 1 and if not, return the index of 0.

Don't try to solve the problem. Your task is only to critique the solution.

Please put your final answer (i.e., the index, which must be 0 or 1) in boxed{{}}. Every output must therefore contain either $\boxed{1}$ or $\boxed{0}$.

---

## L    LIMITATIONS

In this section, we provide rationales behind the datasets and models used in this study.

**Dataset selection**    In section 3, we only use ProcessBench (Zheng et al., 2024a) for evaluating different evaluator implementations (Table 1). ProcessBench provides high-quality human-annotated labels obtained from diverse reasoning traces across 4 benchmarks (GSM8k, MATH, OlympiadBench, and OmniMath) and 12 generators (Llama and Qwen family with varying sizes), suitable for testing different evaluator implementations. While there are other datasets with similar purposes, we resort to ProcessBench for the following reasons:

- Datasets like PRM800k (Lightman et al., 2024), MR-GSM8k (Zeng et al., 2023), and MR-MATH (Xia et al., 2025) annotate errors from GSM8k and MATH, which is already covered by ProcessBench.

- While PRMBench (Song et al., 2025) introduces diverse error types in the MATH dataset, the dataset is generated by synthetically perturbing responses based on a manually developed taxonomy of reasoning errors. Therefore, we make use of ProcessBench that annotates naturally occurring errors in LLM-generated responses without any modification, which better suits the research question *"Can we leverage evaluation-time scaling to improve the generator's performance?"*.

- REVEAL (Jacovi et al., 2024) annotates errors in LLM responses in commonsense reasoning benchmarks, which are not covered by ProcessBench. However, the inter-annotator agreement is significantly lower than ProcessBench ($\kappa \sim 0.5$), indicating high variance in the data label.

**Model selection**    We provide the full list of models in subsection A.1 and subsection A.2. While this paper includes results from a large variety of critic models and direct evaluators, we could not test on other highly capable models due to their recency (Team et al., 2025; Abdin et al., 2025), excessive hardware requirements (using $\sim$70B-sized models as reasoning evaluators), and limited budget for closed-source APIs (*e.g.*, OpenAI o1, Gemini 2.5, Claude 3).

**Task Coverage**    Another limitation of our work is that we do not assess reasoning evaluators for tasks outside math and code. We focus on math and code because (1) they are tasks that admit easy verification and (2) they align with the main strengths of reasoning models. There nonetheless exist other interesting and relevant tasks, including tasks with non-verifiable outputs such as creative and scientific writing. We encourage the research community to extend our work to include such tasks.

## M    LICENSES

### M.1    DATASETS

We disclose the licenses of the datasets used in this study, as indicated in their official HuggingFace repository (if applicable).

- ProcessBench: Apache 2.0
- GSM8k: MIT
- MATH: MIT
- AIME24: CC0 1.0
- AMC23: License type not mentioned. Copyright © Mathematical Association of America.
- MinervaMath: MIT (LM-eval-harness)
- OlympiadBench: Apache 2.0
- LeetCode: MIT
- GPQA: MIT

## M.2 MODELS

We disclose the licenses of the model used in this study, as indicated in their official HuggingFace repository.

- Llama 3 family and their derivatives: Meta Llama 3 Community License. Copyright © Meta Platforms, Inc. All Rights Reserved.
- Qwen family and their derivatives: Qwen License. Copyright © Alibaba Cloud. All Rights Reserved.
- QwQ family: Apache 2.0
- DeepSeek-R1 family: MIT
- Skywork family: Skywork Community License
- Prometheus 2 family: Apache 2.0

# N COMPUTE RESOURCES

To conduct our experiments with 7B and 32B models we used as reasoning evaluators, we used a single node of A6000 GPUs (8 GPUs) where each GPU has 48GB of memory.

# O BROADER IMPLICATIONS

One fundamental issue with language model evaluators is that they were developed to automate human evaluation, which is expensive and time-consuming. However, verifying that these evaluators function as intended, even partially, remains crucial. In our paper, we only tested benchmarks related to mathematics and coding, but this verification becomes even more important when applying reasoning evaluators to assess language model responses on socially sensitive real-world queries or in safety-related domains. We hope that all researchers and practitioners using LM evaluators will consider these points.

# P USE OF LARGE LANGUAGE MODELS

We have used LLMs for writing this paper. Specifically, we have used it to fix grammar and enhance fluency.

