# OpenReview forum: "Scaling Evaluation-time Compute with Reasoning Models as Process Evaluators"
_ICLR.cc/2026/Conference — ICLR 2026 Conference Withdrawn Submission_

### Official Review · Reviewer_k9qW · 2025-10-26

**Soundness:** 2
**Presentation:** 2
**Contribution:** 2
**Rating:** 4
**Confidence:** 4

**Summary:**

The authors explore scaling evaluation-time compute by employing reasoning models to assess both the overall candidate response and the individual reasoning steps within it. In their experiments, they observe that evaluator performance improves monotonically with the number of reasoning tokens generated, mirroring trends observed in LM reasoning. Furthermore, they utilize these more accurate evaluators to rerank multiple generations and demonstrate that allocating more compute during evaluation can be as effective as increasing compute during generation for enhancing an LM's problem-solving performance.

**Strengths:**

The authors present their methodology with clarity, supported by diagrams, and the results effectively enhance the accuracy of the LLM's direct outputs.

**Weaknesses:**

1. The authors' methodology appears somewhat oversimplified, merely combining PRM and ORM in a straightforward manner, which raises doubts about the innovation of this approach.

2. The method proposed by the authors has not been extensively validated in experiments; the version mentioned in the experiments is only the simplified one described in Section 2.2.

3. There is no direct comparison of the computational cost between the baselines and proposed methods. It is recommended to at least provide the testing time and/or the number of input-output tokens used.

4. I have concerns about the experimental setup for "Matching test-time budget across methods." The transparency seems insufficient. The authors claim to use the same test-time budget, but under their method, which involves a total of N*M calls, what are the actual values of N and M in the setup? How was the test-time budget controlled to be the same? Was the comparison method simply repeated multiple times? If so, this does not seem like a fair comparison. If not, how was the comparison conducted?

5. The experimental conclusions in Section 3.2 are all quite straightforward and do not appear to reveal any surprising findings.

**Questions:**

Same as above

---

### Official Review · Reviewer_uct6 · 2025-11-01

**Soundness:** 2
**Presentation:** 3
**Contribution:** 2
**Rating:** 4
**Confidence:** 3

**Summary:**

This paper conducts a systematic study of the use and test-time scaling pattern of reasoning language models as process evaluators, i.e., assessing not only the final answer (outcome evaluation) but also the reasoning trace (process evaluation)—to improve the effectiveness of Best-of-N evaluation accuracy and other inference-time scaling strategies. Through empirical experiments on mathematical reasoning and coding benchmarks, they demonstrate that reasoning models used as process+outcome evaluators can outperform strong baselines (including large ORM/PRM evaluators) even with fewer candidates (8 vs. 64).

**Strengths:**

- The paper studies an important problem: the test-time scaling of process evaluators. While test-time scaling of reasoning itself has been heavily studied, the behavior for evaluators has been underexplored, and this paper provides a comprehensive study.
- The experiments are thorough and strong. Combining multi-step process evaluation is shown to bring consistent gain to single-step process evaluation across reasoning models used. Reasoning outcome evaluator and process evaluator show between scaling trend than direct outcome and process evaluators.
- Insightful analysis: The qualitative analysis in Figure 4 shows that the reasoning evaluator introduces complex behaviors like backtracking, verification, and edge-case thinking during the evaluation process itself.

**Weaknesses:**

- Although the paper examines many method combinations, it is unclear what the defined main method is. The core narrative focuses on "multi-step process evaluation". However, Table 2 results show that the "Reasoning Outcome Evaluator" (51.1) performs slightly better than the "Reasoning Process Evaluator" (50.3). The best result (52.0) comes from a combined "Process + Outcome" method.
- Passive/Inflexible Compute Scaling Mechanism: The paper's method for scaling compute is structural and rigid. Compute is increased N-fold by evaluating N steps, one pass per step, unlike other active test-time scaling methods such as employing budget forcing like "wait" tokens for reasoning continuation.
- While the paper shows promising performance of evaluators in inference time, it is not combined with actual generator models for actual RL training, to see if better process evaluation brings better reasoning training.
- SOTA proprietary API models are not compared in the main paper. It is briefly revealed in Table 4 of Appendix that the best performance of the paper largely underperforms O1-mini in a single-step setting, and there is no multi-step experiment with O1-mini.

**Questions:**

See above.

---

### Official Review · Reviewer_e24F · 2025-11-03

**Soundness:** 2
**Presentation:** 2
**Contribution:** 1
**Rating:** 2
**Confidence:** 4

**Summary:**

This paper explores the use of reasoning models as process evaluators, proposing a long-CoT paradigm to improve trajectory selection in Best-of-N sampling. While the empirical results demonstrating improved task performance are valuable, the paper's novelty and experimental rigor could be strengthened to more firmly establish its contribution.

**Strengths:**

- The proposed the evaluation method is straightforward and show improving task performance.

**Weaknesses:**

- **Clarity of Contribution**: The paper clearly demonstrates that a long-CoT evaluator can enhance performance over direct evaluation. However, the observation that extended reasoning improves LLM-based evaluation, and that process-based evaluation outperforms outcome-based evaluation, has been established in recent literature. The paper would benefit from a more precise delineation of how the proposed "long-CoT" method constitutes a distinct advance beyond simply applying more reasoning steps, which is a known and explored direction.


- **Experimental Controls**: The experimental setup requires tighter controls to validate the conclusions robustly.

  - In Section 3.1, the direct evaluators and reasoning evaluators are based on models with different post-training data. This confounds the effect of the evaluation method with the inherent capabilities of the underlying models. A comparison using the same base model family would isolate the benefit of the long-CoT approach more clearly. Including more contemporary, SOTA models as baselines would further strengthen the results.

  - Similarly, in Section 4.1 (Figure 3), the models used for direct and process-based evaluation during Best-of-N sampling are different. To ensure a fair comparison, the same state-of-the-art model should be used for both evaluation conditions.

**Questions:**

N/a

---

### Official Review · Reviewer_HnE6 · 2025-11-04

**Soundness:** 1
**Presentation:** 3
**Contribution:** 1
**Rating:** 2
**Confidence:** 5

**Summary:**

The research proposes that by using reasoning models as process evaluators and forcing them to spend more compute during evaluation (Evaluation-Time Compute) to generate long Chain-of-Thought, evaluator and generator performance can be improved, demonstrating this "deep evaluation" strategy is superior to the traditional "more generation" (e.g., Best-of-N) strategy under a fixed total compute budget.

**Strengths:**

1. The paper uniquely frames the problem as a systematic trade-off between scaling Evaluation-Time Compute and Generation-Time Compute, providing a new direction for improving LLM performance.

**Weaknesses:**

1.  **Model Obsolescence:** The models used (e.g., Llama-3.1, Qwen2.5, DeepSeek-R1) are from an older generation, making the experimental results potentially non-representative for an ICLR 2026 submission.
2.  **Unverified Premise on Token Scaling:** The central assumption that increasing reasoning tokens monotonically improves evaluator performance lacks validation against models optimized for efficiency. The authors did not train an RM using RL on a small dataset to demonstrate that effective reasoning tokens often *decrease* with training, which would directly challenge the paper's main hypothesis of needing more tokens.
3.  **Empirical Score Aggregation:** The design of the final aggregate score ($s_{final}$) using the interpolation ratio $\alpha$ (e.g., $\alpha=0.5$) is highly empirical and lacks a strong theoretical or principled justification. This dependence on heuristics compromises the robustness of the reported performance gains.
4.  **Lack of Fundamental Novelty:** The methodology is largely an assembly of existing techniques, including LM-as-a-judge, CoT evaluation, and PRMs. The fundamental novelty of the underlying components is thus limited.

**Questions:**

The paper's core premise, which claims that evaluator performance improves monotonically with the number of reasoning tokens generated, is contradicted by the widely observed community trend. Numerous papers leveraging RL (such as PPO or GRPO) on specific datasets report that as models become more optimized, the number of tokens required for accurate evaluation often significantly decreases. The paper fails to validate its approach against an RL-trained RM to verify this critical trade-off.

---

### Official Review · Reviewer_BnVU · 2025-11-07

**Soundness:** 3
**Presentation:** 3
**Contribution:** 2
**Rating:** 4
**Confidence:** 3

**Summary:**

This paper investigates scaling test-time compute for evaluation by using reasoning models as evaluators. The authors propose combining outcome evaluation (assessing final answers) and process evaluation (assessing each reasoning step). Key findings: (1) A 32B reasoning evaluator outperforms a 72B trained PRM by 4.5% on ProcessBench without additional training; (2) In Best-of-N sampling, reasoning evaluators with N=8 outperform direct evaluators with N=64 within comparable compute budgets. Experiments cover 7 benchmarks (mostly math), multiple model families, and include preliminary self-evaluation analysis.

**Strengths:**

1.Demonstrates that evaluation-time compute scaling improves performance analogously to generation, with reasoning evaluators matching trained PRMs without supervision—valuable for domains lacking process labels.

2.Well-structured with clear motivation, logical flow, and accessible writing.

3.Comprehensive experiments across multiple models/benchmarks with thorough ablations.

**Weaknesses:**

1.The novelty is limited because generative evaluation is inherently a generation task, so it is unsurprising that test-time scaling laws applicable to generation also apply here.

2.In Section 3.2's Finding 2, the comparison is biased because single-step evaluators are untrained prompted models while you compare against them. Training generative evaluators like DeepseeGRM (Zijun Liu, et al. 2025) might yield different conclusions

3.Figure 3 shows Reasoning Process Evaluators consume ~10× higher compute than Outcome Evaluators yet achieve lower Best-of-N performance, contradicting the paper's premise that more evaluation-time compute improves results—no satisfactory explanation provided.

---

**Reference**

1.[Zijun Liu, et al. 2025] Inference-Time Scaling for Generalist Reward Modeling

**Questions:**

see **Weaknesses**

---

### Note · Authors · 2025-11-25

I have read and agree with the venue's withdrawal policy on behalf of myself and my co-authors.